# A bispecific antibody exhibits broad neutralization against SARS-CoV-2 Omicron variants XBB.1.16, BQ.1.1 and sarbecoviruses

Yingdan Wang[1,2,6], Aihua Hao[1,6], Ping Ji[1,6], Yunping Ma[1,3,6], Zhaoyong Zhang[4,6], Jiali Chen[1], Qiyu Mao[1], Xinyi Xiong[4], Palizhati Rehati[1], Yajie Wang[1], Yanqun Wang[4], Yumei Wen[1], Lu Lu[1], Zhenguo Chen[1], Jincun Zhao[4,5]✉, Fan Wu[3]✉, Jinghe Huang[1,2]✉ & Lei Sun[1]✉

The Omicron subvariants BQ.1.1, XBB.1.5, and XBB.1.16 of SARS-CoV-2 are known for their adeptness at evading immune responses. Here, we isolate a neutralizing antibody, 7F3, with the capacity to neutralize all tested SARS-CoV-2 variants, including BQ.1.1, XBB.1.5, and XBB.1.16. 7F3 targets the receptor-binding motif (RBM) region and exhibits broad binding to a panel of 37 RBD mutant proteins. We develop the IgG-like bispecific antibody G7-Fc using 7F3 and the cross-neutralizing antibody GW01. G7-Fc demonstrates robust neutralizing activity against all 28 tested SARS-CoV-2 variants and sarbecoviruses, providing potent prophylaxis and therapeutic efficacy against XBB.1 infection in both K18-ACE and BALB/c female mice. Cryo-EM structure analysis of the G7-Fc in complex with the Omicron XBB spike (S) trimer reveals a trimer-dimer conformation, with G7-Fc synergistically targeting two distinct RBD epitopes and blocking ACE2 binding. Comparative analysis of 7F3 and LY-CoV1404 epitopes highlights a distinct and highly conserved epitope in the RBM region bound by 7F3, facilitating neutralization of the immune-evasive Omicron variant XBB.1.16. G7-Fc holds promise as a potential prophylactic countermeasure against SARS-CoV-2, particularly against circulating and emerging variants.

Continuously, sub-lineages of Omicron spread across the world since Omicron breakthrough in Nov. 2021, with increased transmissibility[1] and resistance to humoral immunity of convalescent and vaccine[2,3]. BQ.1.1, XBB.1.5, and XBB.1.16 have been reported to have immune evasion greater than those of earlier sub-lineages of Omicron[1,4,5]. BQ.1.1 has evolved from the BA.5 sub-lineage with three additional mutations

(R346T, K444T, and N460K) on the receptor-binding domain (RBD) of spike (S) protein. XBB.1.5 is a recombinant of the Omicron sub-lineages of BA.2.75 and BJ.1 with five more RBD substitutions (R346T, L368I, V445P, F486P, and F490S) than BA.2.75[6–8]. XBB.1.16, another descendent lineage of XBB with two additional mutations (E180A, T478R) on the spike compared to XBB.1.5, has been spread rapidly

[1]Key Laboratory of Medical Molecular Virology (MOE/NHC/CAMS), Shanghai Institute of Infectious Disease and Biosecurity, Shanghai Fifth People's Hospital, Institutes of Biomedical Sciences, School of Basic Medical Sciences, Fudan University, Shanghai, China. [2]Shanghai Frontiers Science Center of Pathogenic Microorganisms and Infection, School of Basic Medical Sciences, Fudan University, Shanghai, China. [3]Shanghai Immune Therapy Institute, Shanghai Jiao Tong University School of Medicine Affiliated Renji Hospital, Shanghai, China. [4]State Key Laboratory of Respiratory Disease, National Clinical Research Center for Respiratory Disease, Guangzhou Institute of Respiratory Health, the First Affiliated Hospital of Guangzhou Medical University, Guangzhou, Guangdong, China. [5]Shanghai Institute for Advanced Immunochemical Studies, School of Life Science and Technology, ShanghaiTech University, Shanghai, China. [6]These authors contributed equally: Yingdan Wang, Aihua Hao, Ping Ji, Yunping Ma, Zhaoyong Zhang. ✉e-mail: zhaojincun@gird.cn; wufan@fudan.edu.cn; Jinghehuang@fudan.edu.cn; llsun@fudan.edu.cn

in 31 countries[9]. Due to the additional RBD mutations, BQ.1.1, XBB.1.5, and XBB.1.16 became resistant to a wide range of mAbs and antibody cocktails, including those authorized for emergency clinical use, such as LY-CoV1404 (also known as bebtelovimab) and Evusheld (also known as tixagevimab and cilgavimab cocktail)[2,4,10,11].

Thousands of neutralizing antibodies targeting different epitopes of the spike protein have been identified and characterized[12–19]. The majority of these antibodies recognize the RBD of the spike protein, while a small subset targets the NTD, SD1, SD2, or S2 stem-helix[20]. S3H3 antibody[21] targeting SD1 as well as 12-16 and 12-19 antibodies[22] targeting the NTD-SD1 were able to neutralize XBB.1.5 and BQ.1.1. Antibodies targeting RBD have proven effective in neutralizing SARS-CoV-2. However, due to the viral evolution pressure and high genetic variability of the RBD, most of the RBD antibodies failed to neutralize emerging SARS-CoV-2 variants of concern (VOCs), such as BQ.1.1, XBB.1.5, and XBB.1.16. Only the cross-neutralizing antibodies targeting the conserved epitopes of sarbecoviruses in RBD, such as S309, SA55, and S2K146, remained effective for current SARS-CoV-2 VOCs[10,11]. The S2 subunit is highly conserved, and the neutralization activities of S2 stem-helix antibodies were less affected by the viral escape mutation[23]. However, clinical application is unfortunately limited because of its low potency.

With the emergence and rapid dissemination of SARS-CoV-2 variants, the imperative for broadly neutralizing antibodies capable of pan-sarbecovirus neutralization has intensified. Antibodies possessing extensive neutralization across various sarbecoviruses are more likely to target conserved epitopes, rendering them more resilient against immune evasion by swiftly emerging SARS-CoV-2 variants. Consequently, the development of novel pan-sarbecovirus antibodies exhibiting broad and potent activity has become an urgent necessity for both the prevention and treatment of coronavirus disease 2019 (COVID-19). Moreover, considering that coronaviruses have incited two pandemics and one epidemic within the past two decades involving SARS-CoV, SARS-CoV-2, and MERS-CoV, the demand for pan-sarbecovirus antibodies becomes paramount in anticipation of potential recurrences of coronavirus pandemics. Here, we identified an RBM-binding antibody, designated 7F3, and demonstrated its unique capability to neutralize various SARS-CoV-2 VOCs, including the newly emerging Omicron subvariants BQ.1.1, XBB.1.5, and XBB.1.16. Importantly, G7-Fc, a bispecific antibody incorporating GW01 and 7F3 antibodies, showed broad neutralization of SARS-CoV-2 variants and sarbecoviruses reported. G7-Fc prevented XBB.1 infection in mice and bound two RBD sites: 7F3 bound a conserved epitope within the RBM, and GW01 bound an epitope external to the RBM. These studies highlight a conserved epitope within the RBM region targeted by 7F3 that mediated the neutralization of the highly immune-evasive Omicron variants BQ.1.1, XBB.1.5, and XBB.1.16. G7-Fc may represent a potential prophylactic countermeasure against SARS-CoV-2.

## Results

### Isolation of a BQ.1.1 and XBB lineages neutralizing antibody 7F3 from a COVID-19 convalescent individual

We sorted and cultured memory B cells from a patient who had recovered from COVID-19 (Supplementary Fig. S1). After culture of the B cells for two weeks, we screened the supernatants of B cells for SARS-CoV-2 neutralization. We identified a SARS-CoV-2 neutralizing antibody (NAb), designated 7F3, which belongs to the IGHV3-49 and IGKV3-11 immunoglobulin gene families. The antibody 7F3 exhibited potent binding to the RBD proteins of the SARS-CoV-2 (Wuhan-Hu-1 strain, GenBank: MN908947.3) wild-type (WT), Delta, BQ.1.1, XBB, XBB.1.5, and XBB.1.16 variants, and weak binding to the BA.1 and BA.5 RBDs, as measured by enzyme-linked immunosorbent assay (ELISA, Fig. 1a). In contrast, the control broadly neutralizing monoclonal antibody LY-CoV1404 bound robustly only to the RBDs of wild-type (WT), Delta, BA.1, and BA.5 variants but did not engage the BQ.1.1,

XBB, XBB.1.5, or XBB.1.16 RBDs. We previously isolated a broadly NAb from a COVID-19 convalescent, named GW01[24], which targeted the outside of RBM and displayed broadly neutralization potency against SARS-CoV-2 and SARS-CoV. However, it failed to bind BQ.1.1, XBB (Fig. 1a). The binding affinities of 7F3, GW01, and LY-CoV1404 for the RBDs of different SARS-CoV-2 strains accorded with the results of ELISA (Fig. 1b, Supplementary Fig. S2, and Supplementary Table S1).

To assess the potency and breadth of 7F3, we conducted neutralization tests against a panel of 15 pseudotyped viruses expressing the spike region of sarbecoviruses. Sarbecoviruses, encompassing a group within the Betacoronavirus genus, share genetic affinities with the SARS-CoV and exhibit commonalities with other coronaviruses. Specifically, non-SARS-CoV-2 viruses, including SARS-CoV (SARS1), RS3367, and WIV1, are categorized as clade 1a sarbecoviruses, whereas SARS-CoV-2 variants fall under clade 1b sarbecoviruses. 7F3 displayed a notable absence of neutralization activity against clade 1a sarbecoviruses. However, it demonstrated potent neutralization against clade 1b sarbecoviruses, including SARS-CoV-2, Alpha, Beta, and Delta at sub-nanomolar level. Moreover, 7F3 exhibited broad spectrum neutralization capacity against all the tested Omicron subvariants, including BA.1, BA.5, BF.7, and the highly resistant variants BQ.1.1, XBB.1.5, XBB.1.16, and XBB.1.16.1 (Fig. 1c). In comparison, LY-CoV1404 exhibited high neutralization potency against various SARS-CoV-2 strains with the exception of the Omicron subvariants BQ.1.1, XBB, XBB.1.5, XBB.1.16, and XBB.1.16.1 (Fig. 1c). GW01 exhibited high neutralization potency against the clade 1a sarbecoviruses, such as SARS-CoV, RS3367, and WIV1. It also neutralized various clade 1b SARS-CoV-2 variants except for the Omicron subvariants (Fig. 1c). An examination of the combined breadth and potency of 7F3 and GW01 in a 1:1 ratio revealed no discernible enhancement in neutralizing SARS-CoV-2 strains and Omicron subvariants (Fig. 1c). These results suggest that 7F3 is a promising candidate for the development of broadly neutralizing antibody therapies against SARS-CoV-2, particularly in the context of the current surge in Omicron subvariants.

### 7F3 blocks ACE2 binding and exhibits a robust ability to bind RBD mutants

In a bilayer interferometry (BLI) competition assay, 7F3, LY-CoV1404, and GW01 impeded binding of the wild-type SARS-CoV-2 spike RBD to angiotensin-converting enzyme 2 (ACE2) (Fig. 1d). In contrast, the control antibody S309, which is a class III antibody and recognizes an epitope outside the receptor-binding motif (RBM), exhibited no competition. These data indicate that 7F3 might bind within the RBM region or sterically block ACE2 from binding to the RBM.

To map the epitope of 7F3 within the RBD, we used a panel of 37 RBD mutant proteins bearing the circulating single or triple amino acid mutations and evaluated 7F3 binding to these mutants compared to the wild-type RBD. 7F3 retained binding to 100% of the tested RBD mutants (Fig. 1e). In contrast, antibodies LY-CoV1404 and GW01 displayed diminished binding (below 85% of binding to the Wuhan-Hu-1 strain) to 9 and 5 RBD single mutants, respectively, and most RBD triple mutants (Fig. 1e). These findings suggest that 7F3 targets an epitope within RBD that distinguishes it from other characterized antibodies we investigated, which may enable its broad and potent activity against circulating and emerging SARS-CoV-2 variants.

### Construction and binding ability of bispecific antibody G7-Fc

We previously reported that IgG-like bispecific antibodies constructed by non-Omicron neutralizing antibodies were able to neutralize the Omicron variants BA.1 and BA.2[24,25]. Therefore, we constructed bispecific antibodies using 7F3 and GW01 to see if these two antibodies in IgG-like structure format could enhance the breadth and potency of 7F3. Two bispecific antibodies, GW01-7F3-Fc (G7-Fc) and 7F3-GW01-Fc (7G-Fc), were successfully created as described presviously[25] (Fig. 2a). Briefly, the single-chain variable fragments (scFv) of antibodies were

linked with a (Gly4Ser)4 linker, and subsequently fused to IgG1 Fc. The sequential arrangement of antibody G7-Fc, from N to C terminus, is delineated as follows: GW01 VL-(Gly4Ser)3-GW01 VH-(Gly4Ser)4-7F3 VL-(Gly4Ser)3-7F3 VH-Hinge-CH2-CH3. In contrast, the 7G-Fc

configuration positions the single chain of 7F3 in the N terminal, followed by the single chain of GW01. The formation of disulfide bonds within the hinge region imparts a dimeric structure to the bispecific antibody. The single chain of bispecific antibodies was approximately

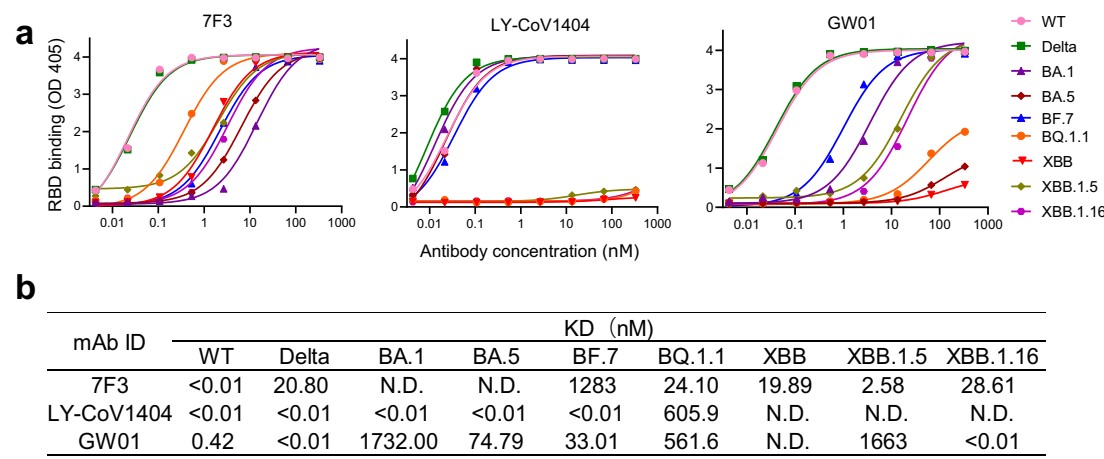

**Fig. 1 | Isolation of BQ.1.1 and XBB.1.5 neutralizing antibody 7F3 from COVID-19 convalescent individuals. a** Binding specificity of 7F3 to the RBDs of SARS-CoV-2, Delta, BA.1, BA.5, BF.7, BQ.1.1, XBB, XBB.1.5, and XBB.1.16 as measured in an ELISA. LY-CoV1404 and GW01 were used as controls. **b** Binding affinity KD values of 7F3 to the receptor binding domains (RBDs) of SARS-CoV-2 variants as measured by BLI. LY-CoV1404 and GW01 were used as controls. **c** Neutralizing activities of 7F3, LY-CoV1404, GW01, and the combination of 7F3 and GW01 at the ratio of 1:1 against pseudotyped SARS-CoV-2 subvariants as well as sarbecoviruses. The number displayed in the box represents the half-maximal inhibitory concentration (IC50)

values. **d** Binding of 7F3 to the SARS-CoV-2 RBD in competition with ACE2 as measured by BLI experiments. The HIV-1 antibody VRC01, severed as an IgG1 isotype-matched negative control, and a mixture of VRC01 and the ACE2 receptor served as a positive control. Blockage was calculated by the values of maximum binding. The maximum binding of positive control (ACE2) was set as 100%, and negative control was set as 0%. **e** Binding of 7F3 to 33 RBD-Fc proteins containing single or triple mutations. GW01 and LY-CoV1404 were used as controls. The experiments in (**c**) were independently performed twice with similar results. Source data are provided as a Source Data file.

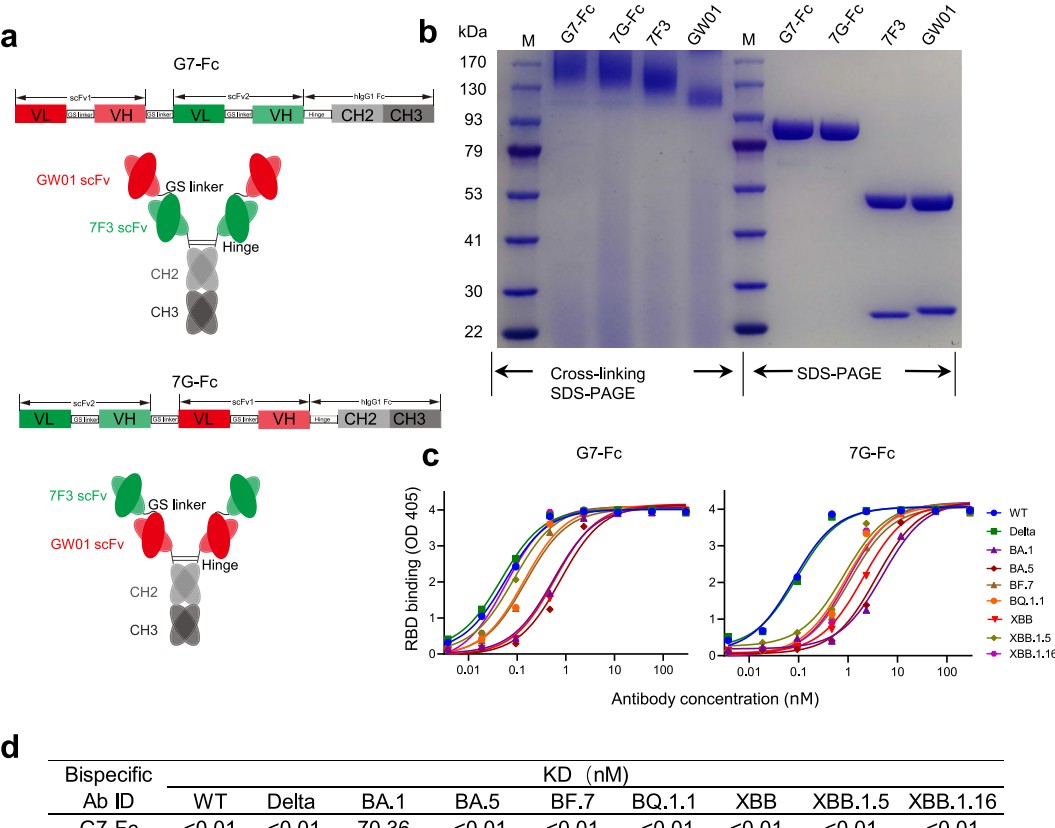

**d**

| Bispecific Ab ID | KD (nM) | | | | | | | | |
|---|---|---|---|---|---|---|---|---|---|
| | WT | Delta | BA.1 | BA.5 | BF.7 | BQ.1.1 | XBB | XBB.1.5 | XBB.1.16 |
| G7-Fc | <0.01 | <0.01 | 70.36 | <0.01 | <0.01 | <0.01 | <0.01 | <0.01 | <0.01 |
| 7G-Fc | <0.01 | <0.01 | 2598 | <0.01 | 53.70 | <0.01 | 9.18 | 213.00 | <0.01 |

**e**

| Virus ID | IC50 (nM) | | | | | |
|---|---|---|---|---|---|---|
| | G7-Fc | 7G-Fc | LY-CoV1404 | 7F3 | GW01 | 7F3+GW01 |
| WT | 0.1017 | 0.1908 | 0.00874 | 0.01781 | 0.2159 | 0.3801 |
| B.1.1.7 | 0.1178 | 0.3116 | 0.03898 | 0.04163 | 0.2424 | 0.1733 |
| B.1.351 | 0.2526 | 0.1631 | 0.00758 | 0.03436 | 0.2093 | 0.1554 |
| B.1.617.2 | 0.1428 | 0.0813 | 0.01944 | 0.02752 | 0.1502 | 0.2779 |
| BA.1 | 0.4169 | 8.863 | 0.05272 | 104.3 | >333.3 | 126.9 |
| BA.2.12.1 | 0.03566 | >294.11 | 0.01765 | 168.8 | >333.3 | 90.1 |
| BA.2.75 | 0.1181 | 0.0932 | 0.02630 | 0.3026 | >333.3 | 0.773 |
| BL.1 | 0.2129 | 0.5150 | 0.05724 | 1.082 | >333.3 | 1.825 |
| BA.2.75.2 | 0.0798 | 34.520 | 0.1499 | 146.7 | >333.3 | 15.060 |
| BN.2.1 | 0.04209 | 0.2234 | 7.356 | 3.521 | >333.3 | 1.964 |
| CA.1 | 0.4206 | 34.470 | 2.848 | >333.3 | >333.3 | 242.5 |
| BA.2.75.4 | 0.0978 | 0.2216 | 1.213 | 0.1860 | >333.3 | 0.6051 |
| BR.1 | 0.1343 | 0.2195 | 11.290 | 0.5837 | >333.3 | 0.859 |
| CH.1.1 | 0.2022 | >294.11 | 6.368 | >333.3 | >333.3 | >333.3 |
| XBB | 0.3206 | 183.1 | >333.3 | 44.280 | >333.3 | 65.630 |
| XBB.1.5 | 0.2173 | 2.073 | >333.3 | 0.786 | >333.3 | 1.081 |
| XBB.1.16 | 0.4263 | 51.300 | >333.3 | 16.080 | >333.3 | 14.910 |
| XBB.1.16.1 | 0.3281 | >294.11 | >333.3 | 21.210 | >333.3 | 50.260 |
| BA.5 | 0.03818 | 19.520 | 0.03098 | 62.520 | 276.1 | 20.910 |
| BF.7 | 0.02102 | 1.231 | 0.03972 | 23.130 | 242.5 | 40.070 |
| BA.4.6 | 0.1428 | >294.11 | 0.04879 | >333.3 | >333.3 | 243.1 |
| BA.5.1.12 | 0.3296 | >294.11 | 3.288 | >333.3 | >333.3 | >333.3 |
| BA.5.6.2 | 0.2283 | >294.11 | >333.3 | >333.3 | >333.3 | >333.3 |
| BU.1 | 0.2465 | 27.910 | 1.611 | 13.040 | >333.3 | 48.890 |
| BQ.1.1 | 0.1143 | 56.240 | >333.3 | 2.054 | >333.3 | 4.778 |
| SARS-CoV | 0.2484 | 0.758 | >333.3 | >333.3 | 0.843 | 2.719 |
| RS3367 | 0.03830 | 0.03344 | >333.3 | >333.3 | 0.1042 | 0.1864 |
| WIV1 | 0.02897 | 0.01298 | >333.3 | >333.3 | 0.02446 | 0.01878 |
| % of Neut | 100.0 | 78.6 | 67.9 | 71.4 | 32.1 | 89.3 |
| GM IC50 | 0.13 | 1.37 | 0.18 | 2.52 | 0.85 | 4.21 |

| <333.3 | <66.6 | <6.66 | <0.66 | <0.066 |
|---|---|---|---|---|

85 kDa, while the full-length crosslinked bispecific antibodies were about 170 kDa after glutaraldehyde-mediated crosslinking, as visualized by SDS-PAGE (Fig. 2b). This mass is only about 10–20% larger than the full-size IgG of 7F3 (150 kDa) and GW01 (130 kDa).

Bispecific antibody G7-Fc strongly bound to the RBDs of Wuhan-Hu-1 (WT), Delta, BA.1, BA.5, BQ.1.1, XBB.1.5, and XBB.1.16 (Fig. 2c) and exhibited a higher binding affinity towards these proteins

compared to the parental antibodies (Fig. 2d, Supplementary Fig. S3, and Supplementary Table. S1). The bispecific antibody 7G-Fc, which has a reverse orientation of GW01 and 7F3, was found to be less efficient in binding Omicron RBD proteins than G7-Fc, with lower binding affinity and bigger equilibrium dissociation constant (KD) values (Fig. 2d). These results suggested that the structure of the bispecific antibody and the order of the parental antibodies impact

**Fig. 2 | Neutralization breadth and potency of bispecific antibody G7-Fc.**
**a** Schematic diagrams of the structure of bispecific antibodies. **b** Cross-linking
sodium dodecyl sulfate-polyacrylamide gel electrophoresis (SDS-PAGE) and SDS-
PAGE analysis of G7-Fc, 7G-Fc, 7F3, GW01. Binding specificities (**c**), binding affinities
KD values (**d**) of the bispecific antibodies G7-Fc and 7G-Fc to the RBD-His proteins
of WT, Delta, BA.1, BA.5, BF.7, BQ.1.1, XBB, XBB.1.5, XBB.1.16 as measured by ELISA
and BLI experiments. ND represents not determined. **e** Neutralization breadth and
potency of bispecific antibodies G7-Fc and 7G-Fc against the SARS-CoV-2 variants

and sarbecoviruses. LY-CoV1404 was used as a control. The number in the box
indicates IC50 values. Geometric mean (GM) IC50 was determined. **f** Neutralization
of authentic viruses BQ.1 and XBB.1 by G7-Fc. The curves were fitted using nonlinear
regression (log [inhibitor] vs. normalized response, variable slope). The dots on the
curve represent the mean values of triplicate experiments, accompanied by bars
indicating the standard error of the mean (SEM). The experiments in **b**, **e** were
independently performed twice with consistent results. Source data are provided in
the Source Data file.

the binding affinity of the bispecific antibody to the RBDs of Omi-
cron subvariants.

## Neutralization activity of bispecific antibody G7-Fc

To understand the breadth and potency of these bispecific anti-
bodies, we performed a neutralization assay using a panel of twenty-
eight pseudoviruses, including SARS-CoV-2 WT (Wuhan-Hu-1 strain),
Alpha, Beta, Delta, twenty-one Omicron subvariants including BQ.1.1,
XBB.1.5, and XBB.1.16, as well as other sarbecoviruses such as SARS-
CoV, WIV1, and RS3367. Remarkably, G7-Fc bispecific antibody
exhibited robust neutralization potency against 100% of the tested
pseudoviruses, with a geometric mean (GM) IC50 value of 0.13 nM
(Fig. 2e), while the parental antibodies GW01, 7F3, and their com-
bination only neutralized 32.1%, 67.9%, and 89.3% with GM IC50
values of 0.85, 2.52, and 4.21 nM, respectively. Significantly, G7-Fc
outperformed the parent IgG and the mixture of IgG. G7-Fc potently
neutralized BQ.1.1, XBB.1.5, and XBB.1.16 with IC50 values of 0.1143,
0.2173, and 0.4263 nM, respectively. The neutralization potency of
7G-Fc, characterized by a reversed orientation of GW01 and 7F3, is
not comparable to that of G7-Fc. This observation corroborates the
notion that the arrangement of parental antibodies significantly
influences both neutralizing activity and binding affinity, as pre-
viously discussed. In contrast, LY-CoV1404 only neutralized 67.9% of
the tested pseudoviruses. Importantly, it failed to neutralize BQ.1.1,
XBB.1.5, XBB.1.16, XBB.1.16.1, and BA.5.6.2 (Fig. 2e). Moreover, G7-Fc
bispecific antibody demonstrated robust neutralization against the
authentic viruses BQ.1 and XBB.1 in vitro (Fig. 2f). Taken together,
these data indicated that bispecific antibody G7-Fc is a sarbecovirus
neutralizing antibody and potently neutralizes the Omicron sub-
variants, especially the highly resistant variants BQ.1.1 and XBB.1.5,
XBB.1.16. These results provide important insights into the devel-
opment of effective treatment options for COVID-19.

## Prophylactic and therapeutic efficacy of G7-Fc against XBB.1 infection of mice

To evaluate the in vivo prophylaxis and therapeutic efficacy of G7-Fc,
we administered the G7-Fc into K18-ACE2 transgenic mice and BALB/c
mice either before or after infection of XBB.1 (Fig. 3a). As XBB.1 tends
to attach to the cells in the upper airways[26], inhalation antibody ther-
apy demonstrated predominant therapeutic efficacy in animal
study[1,24]. To determine the anti-viral activity of G7-Fc by different
administration routes, we administered G7-Fc via intraperitoneal (i.p.)
route (200 µg/mouse) and intranasal (i.n.) route (20 µg/mouse)
(Fig. 3a). Viral load in the lungs of K18-ACE2 transgenic mice was
detected 48 hours post-infection in the control group treated with
phosphate-buffered saline (PBS), with a value of $6.75 \times Log10$ focus-
forming units (FFU)/g. Remarkably, no viruses were detected in any of
the prophylactic or therapeutic groups regardless of administration
routes (i.p. or i.n.) (Fig. 3b). Furthermore, there was no significant
difference between the anti-viral activity of G7-Fc administered using
either i.p. or i.n. delivery routes (Fig. 3b). Similar findings were evident
in BALB/c mice, although a lower lung viral titer of $3.16 \times Log10$ FFU/g
was recorded in the control PBS group (Fig. 3c). Our results demon-
strate that G7-Fc is highly efficacious in both the prevention and
treatment of XBB.1 infection, irrespective of administration route.

## Cryo-EM structure of SARS-CoV-2 Omicron XBB S complexed with G7-Fc

To further understand the neutralization mechanism of G7-Fc, we
determined the cryo-electron microscopy (cryo-EM) structure of the
prefusion-stabilized SARS-CoV-2 Omicron XBB S ectodomain trimer
complexed with G7-Fc in its full-length IgG form (XBB S/G7-Fc) (Sup-
plementary Fig. S4a). Negative staining EM images showed that incu-
bation of G7-Fc with S induced the formation of a trimer dimer
consisting of two S trimers (Supplementary Fig. S4c). However, XBB S
trimers failed to form trimer dimers when incubated with G7-Fab
(Supplementary Fig. S4b, c). The whole complex was determined
to a resolution of 3.75 Å. To obtain detailed information about
the interactions between S and G7-Fc, local refinement focusing on
single trimers and RBD/scFvs was performed and improved the inter-
face region to 3.0 Å (Figs. 4a, e and Fig. 5a, Supplementary
Figs. S5 and S6).

The binding of G7-Fc induces S trimers to the state with all three
RBDs up. Due to the distance limitation of the GS linkers, 7F3 and
GW01 from one arm of the G7-Fc bind to different RBDs, with 7F3
binding to RBM and GW01 binding to a non-RBM region. Two 7F3s are
close to each other but do not form close interactions. The Fc regions
cross-link two S trimers, forming an unsymmetrical head-to-head
dimer of trimers (Fig. 4a, Supplementary Fig. S7). 7F3 did not compete
with GW01 for RBD engagement (0%, Fig. 4b), implying that the 7F3
epitope likely differs from those of GW01 within the RBM. Antibodies
directed towards the RBD can be classified into four general categories
(classes I to IV) based on their competition with the ACE2 and their
recognition of the up or down state of the three RBDs in spike[27].
7F3 strongly competed with the class II antibody BD23[28] for Wuhan-Hu-
1 (WT) RBD engagement (100%, Fig. 4b). In contrast, CR3022 showed
limited competition with 7F3 (17.4%). Moreover, CR3022[29], class IV,
exhibited strong competition to the WT RBD compared to GW01
(96.21%, Fig. 4c), whereas 7F3 and BD23 showed no competition or
limited competition with 7F3 (0% and 34.81%, respectively, Fig. 4c).
These findings suggest a substantial epitope overlap exists between
7F3 and BD23, and overlap between CR3022 and GW01 epitopes.
Based on the Barnes Classification of antibodies, 7F3 is categorized as a
class II antibody, whereas GW01 is classified as a class IV antibody.

7F3 binds to the RBM region and shares 26 epitope residues with
the ACE2 binding site (Fig. 4e, f, Supplementary Fig. S8b). The binding
of 7F3 to RBD buries 976.4 Å² surface area (Supplementary Table S3).
The interactions between 7F3 and RBD are mainly contributed by
CDRH2, CDRH3, and CDRL3 (Fig. 5a). Residues Y449, Y453, L455, F456,
Y473, A475, G476, N477, K478, P479, A484, G485, S486, N487, C488,
Y489, S490, P491, L492, Q493, S494, Y495, G496, R498, Y501, and H505 of RBD participate in the interactions with 7F3, forming 18
pairs of hydrogen bonds (Fig. 5b, Supplementary Table S4). In addi-
tion, Y229 of CDRH3 packs against a hydrophobic pocket in the
interface formed by L455, F456, Y473, and Y489 of RBD (Fig. 5b).
Structure comparison reveals that the binding modes of XBB S/7F3 and
WT S/BD23 (PDB ID: 7BYR) are quite similar. Other than that, 7F3 was
implicated in more interactions with RBD residues, and the E484A,
F486S, Q498R, N501Y, and Y505H mutations may interfere with the
contacts between BD23 and SARS-CoV-2 variants (Supplemen-
tary Fig. S8).

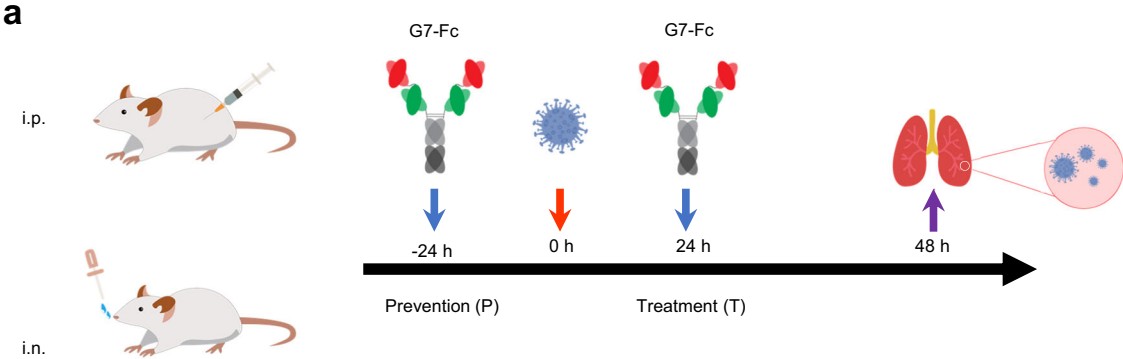

**b**

K18-ACE2
mice

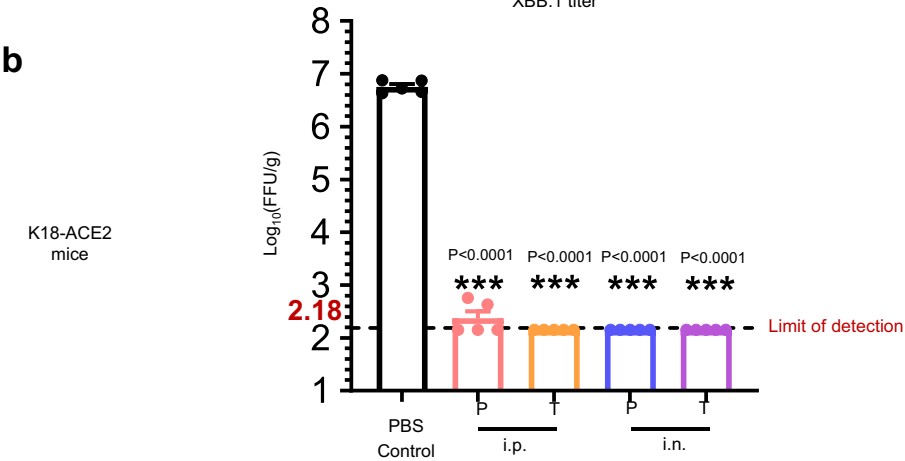

**c**

BALB/c
mice

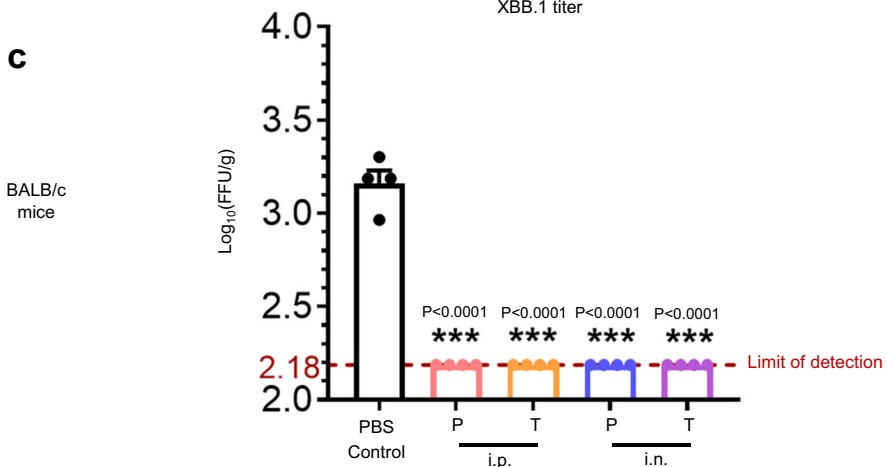

**Fig. 3 | In vivo prophylactic and therapeutic activity of the bispecific antibody G7-Fc via different administration routes. a** Schematic diagram of G7-Fc in the prevention (abbreviation P) and treatment (abbreviation T) of XBB.1 infection. G7-Fc was administrated intraperitoneally (abbreviation i.p.) or intranasally (abbreviation i.n.). **b, c** Viral titers (FFU, focus-forming units) of lung tissue of **b** K8-ACE2 mice (*n* = 5) and **c** BALB/c mice (*n* = 4) were determined after 48 h post-infection.

Each data point represents an individual mouse within the respective groups. The mean ± SEM of all data points are shown in (**b**, **c**). Statistical significance was analyzed using a two-sided *t*-test in (**b**, **c**) with Prism software (version 9, GraphPad Software), and *p* < 0.0001 is expressed as ***. Source data of **b**, **c** are provided in the Source Data file. The icon in **a** were created using Adobe Illustrator, and the schematic diagram was created using Microsoft Office PowerPoint.

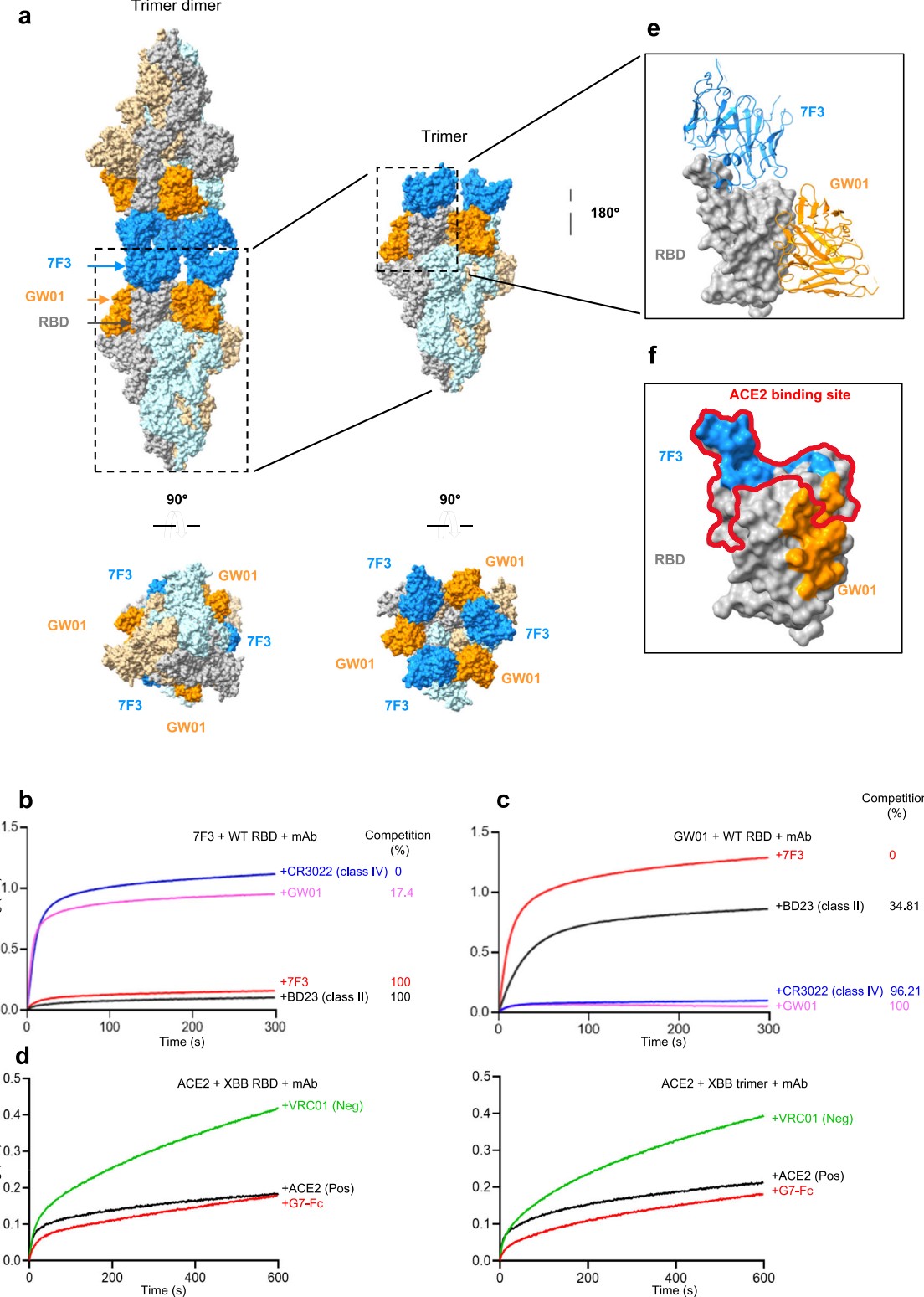

**Fig. 4 | Cryo-EM structures of the XBB S trimer in complex with the bispecific antibody G7-Fc IgG. a** G7-Fc binds to XBB S, forming a trimer dimer. Two perpendicular views of XBB S/G7-Fc depict the surface of the trimer dimer and one of the trimers, with GW01 in orange and 7F3 in dodger blue. **b** Binding of 7F3 to the SARS-CoV-2 RBD in competition with CR3022, GW01, and BD23 was assessed using BLI. **c** Binding of GW01 to the SARS-CoV-2 RBD in competition with 7F3, BD23, and CR3022 was evaluated using BLI. **d** Binding of ACE2 to the RBDs of XBB trimer in competition with bispecific antibody G7-Fc. A control IgG1 was used as a negative control, and the addition of ACE2 was used as a positive control. **e** Close-up view of the interaction between G7-Fc and XBB S. The XBB S-RBD is displayed on a dark gray surface. GW01 and 7F3 are shown as cartoons colored orange and dodger blue, respectively. **f** Surface representation of RBD showing the buried binding site, including GW01, 7F3, and the ACE2 binding site. Source data are provided as a Source Data file.

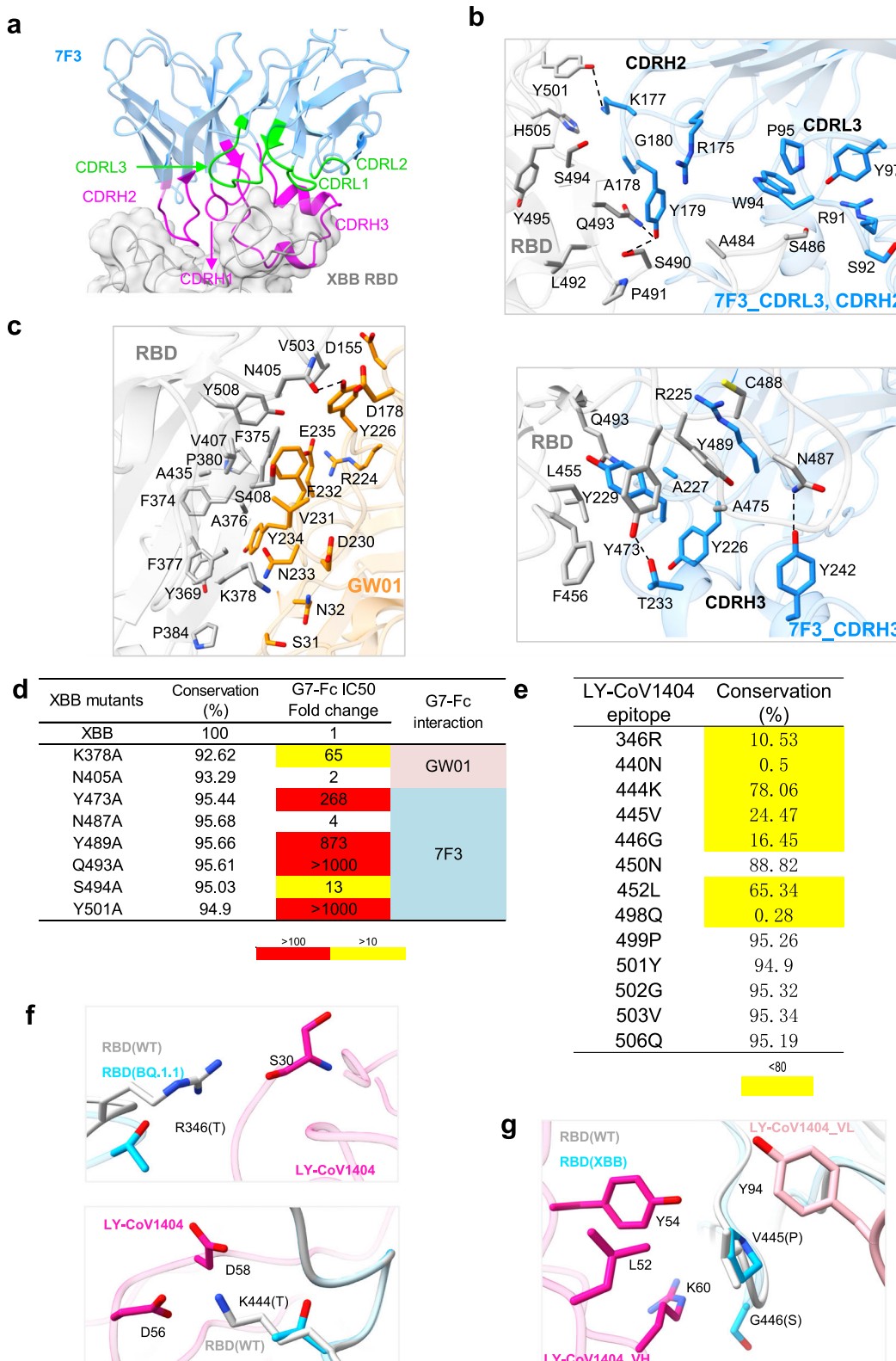

**Fig. 5 | The epitope of G7-Fc on XBB S. a** The interactions between XBB S-RBD and 7F3. **b** The detailed interactions between RBD and 7F3. The residues involved in interactions are represented as sticks. **c** The detailed interactions between RBD and GW01. **d** Mapping the epitope of G7-Fc by site-directed mutagenesis.

**e** Conservation of the epitope of LY-CoV1404 according to the CoV-Spectrum of GISAID database. **f**, **g** The detailed interactions between RBD and LY-CoV1404. Source data are provided as a Source Data file.

The epitope of GW01 is located outside RBM. We previously identified the epitope of GW01 on BA.1 S-RBD[25]. GW01 interacts with BA.1 and XBB S-RBD in a very similar way. The contacts between GW01 and the XBB S-RBD are mainly contributed by CDRH3 by forming hydrophobic interactions and 6 pairs of hydrogen bonds (Supplementary Table S5). Y226, N233, Y234, and E235 of CDRH3 interact with XBB S-RBD through hydrogen bonds. Y226, V231, F232, and Y234 of CDRH3 establish a large hydrophobic interface with Y369, F374, F375, A376, F377, V407, A435, V503, and Y508 of RBD (Fig. 5c). In addition, D155 of CDRH1 and D178 of CDRH2 are also involved in the interactions between GW01 and RBD (Fig. 5c). When compared with BA.1 S-RBD/GW01 structure, fewer residues of XBB S-RBD are involved in the GW01 interaction, resulting in a reduction of interface area by 91.7 $Å^2$ in the XBB S-RBD/GW01 structure (Supplementary Table S3). This may explain why GW01 exhibits lower binding affinities to XBB S-RBD than BA.1 S-RBD. A comparison of XBB S/GW01 and WT S/CR3022 (PDB ID: 6W41) reveals that the CR3022 epitope, which primarily consists of L368-F392 residues, is located near the bottom of the RBD. The S371F, S375F, T376A, and R408S mutations may cause the loss of binding affinity of CR3022 to SARS-CoV-2 variants (Supplementary Fig. S9).

To characterize the key epitope of G7-Fc, we analyzed the residues involved in G7-Fc binding (Supplementary Fig. S8b, 9b) and constructed eight XBB single mutants with crucial roles in G7-Fc binding. The K378A, Y473A, Y489A, Q493A, S494A, and Y501A mutants dramatically decreased (>10 fold and <100 fold) or abolished (>100 fold) G7-Fc neutralization (Fig. 5d). The K378 forms a hydrogen bond with GW01 (Fig. 5c), whereas the Y473, Y489, Q493,S494, and Y501 residues form hydrogen bonds with 7F3. The residues involved in the G7-Fc epitope exhibit a high degree of conservation, particularly with K378, Y473, Y489, Q493, S494, and Y501, showcasing over 95% conservation in variant sequences worldwide since Jan 2023 according to the CoV-Spectrum of GISAID database (https://cov-spectrum.org/explore/World/AllSamples/AllTimes/variants?variantQuery=S%3A466R&) (Fig. 5d, Supplementary Table S6). The conserved binding epitope of 7F3 conferred notable binding breadth to a panel of 37 RBD mutant proteins with single or triple mutations (Fig. 1e). Despite the Y501A mutation leading to escape from G7-Fc neutralization, an analysis of global SARS-CoV-2 genomic databases since January 2023 revealed only 6 reported sequences containing this 501A mutation. The mutation rates of other residues in position Y501 worldwide are notably low (Supplementary Fig. S10b), indicating that escape mutations in position Y501 are uncommon. Importantly, this suggests that G7-Fc can potently neutralize the vast majority of currently circulating SARS-CoV-2 strains. Analysis of the structure of LY-CoV1404 complexed with WT RBD (PDB ID: 7MMO) showed that the epitope of LY-CoV1404 is of low conservation at R346, N440, K444-G446, L452, and Q498 (Supplementary Fig. S10a and Fig. 5e), which explains its decrease of binding affinity of SARS-CoV-2 variants. In detail, the R346T, N440K, K444T, L452R, and Q498R mutations may result in the escape of BQ.1.1 (Fig. 5f). The V445P and G446S disrupt the hydrophobic interface between V445 and L52, Y54, K60 of heavy chain and Y94 of light chain, which may contribute to the escape of XBB, XBB.1.5, and XBB.1.16 (Fig. 5g). These results indicate that G7-Fc bispecific antibody binds to two complementary epitopes across variants of the Omicron lineage of SARS-CoV-2, thereby explaining the ability of G7-Fc to neutralize even the highly antigenically evasive Omicron variants.

### Neutralization mechanism of the bispecific antibody G7-Fc

To investigate the neutralization mechanism of the bispecific antibody G7-Fc, we performed a BLI competition assay and tested whether G7-Fc blocks RBD/ACE2 binding. G7-Fc prevented the RBD protein and the S trimer of XBB binding to ACE2 protein in the competition assay, while the control antibody VRC01, an HIV-1 gp120 binding antibody, did not affect the RBD/ACE2 interaction (Fig. 4d). Structural alignment of the G7-Fc/RBD complex with the ACE2/RBD complex indicated that both 7F3 and GW01 were able to compete with ACE2 when binding to the RBD (Fig. 4f), which was consistent with the competition assay (Fig. 4d).

These findings suggest that G7-Fc inhibits SARS-CoV-2 XBB variant infection by synergistically inducing the formation of trimer-dimers. Both 7F3 and GW01 bind to conserved epitopes in XBB, enlarging the interface area, thereby improving the affinity between the RBD and single scFv and blocking the RBD from interacting with the ACE2 receptor. The structural arrangement of the bispecific antibody, particularly the Fc region, plays a critical role in facilitating G7-Fc binding to the trimer.

## Discussion

The Omicron subvariants BQ.1.1, XBB.1.5, and XBB.1.16, were found to be highly immune evasive variants of severe acute respiratory syndrome coronavirus 2 (SARS-CoV-2). All the antibodies targeting the RBM region, including potent antibody LY-CoV1404, failed to neutralize BQ.1.1, XBB.1.5, and XBB.1.16. In this study, we discovered an RBM-specific antibody 7F3, which neutralized the highly immune-evasive SARS-CoV-2 variants. G7-Fc, incorporating 7F3, showed broad neutralization of variants and sarbecoviruses, were reported and exhibited potent prophylaxis and therapeutic efficacy against XBB.1 infection of mice.

Structural analysis of the XBB trimer bound by G7-Fc revealed that 7F3 interacts with 29 residues in the receptor-binding domain (RBD), forming 18 hydrogen bonds and one hydrophobic interaction patch. These 29 residues localize primarily to the RBM region, including Y449, L455-F456, Y473-K478, and A484-S494. To confirm the unique binding capacity of 7F3, we constructed a series of RBD single mutants and triple mutants. Indeed, 7F3 demonstrated superior binding compared with LY-CoV1404, recognizing all RBD mutants, including successive triple mutants in the RBM region (Fig. 1d). The 7F3 epitope is unique within RBD that distinguishes it from other characterized antibodies. In contrast, the LY-CoV1404 epitope may focus on more variable regions of the RBD that differ substantially between mutants. These findings suggest the notable breadth of G7-Fc can be attributed to the distinctive binding epitope and binding ability of 7F3.

The trimer dimer state of the spike induced by G7-Fc was observed during the structural study of the XBB S/G7-Fc complex. Structural analysis of the XBB S/G7-Fc complex showed that GW01 only binds to the up RBD and would clash with the adjacent down RBD (Supplementary Fig. S7a). Thus, 7F3 binding might induce RBD to the up position, thus facilitating GW01 binding. This resembles the 2-step models for antibody synergy[25,30], where binding of the first antibody induces RBD into the up position and enhances the binding of the second antibody. In addition, when the GW01 in G7-Fc was replaced with other SARS-CoV-2 mAbs, such as 4L12 and REGN10989, the trimer-dimer state of the spike was not induced by incubating 4L12-7F3-Fc or REGN10989-7F3-Fc with XBB S trimer, as observed in negative staining EM images (Supplementary Fig. S4c). These results demonstrate that 7F3 and GW01 work synergistically, inducing the crosslink of spike and improving binding and neutralizing activity with the RBD.

Taken together, our study discovered a specific RBM-binding antibody 7F3, which neutralized the highly immune-evasive SARS-CoV-2 variants BQ.1.1, XBB.1.5, and XBB.1.16. The conserved epitope within the RBM region bound by 7F3 may be critical for receptor binding and viral entry, highlighting it as an attractive target site for vaccine and drug development. Moreover, G7-Fc, incorporating 7F3, exhibited excellent in vivo anti-XBB efficacy in both prevention and treatment therapy via different administration routes. G7-Fc may be used as a potential prophylactic countermeasure against SARS-CoV-2.

## Methods

### Cell lines and participants

HEK293T cells (ATCC, CRL-3216) and Huh-7 cells (National Collection of Authenticated Cell Cultures, CSTR: 19375.09.3101HUMSCSP526) were cultured in Dulbecco's Modified Eagle Medium (DMEM) supplemented with 10% fetal bovine serum (FBS), 1000 units/mL streptomycin, and 1000 µg/mL penicillin at 37 °C in a 5% $CO_2$ atmosphere. HEK293F suspension cells (Thermo Fisher Scientific, R79007) were cultured in serum-free SMM 293-TII Expression Medium (Sino Biological Inc.) at 37 °C in a 5% $CO_2$ atmosphere with shaking at 120 rpm. Ethical approval of this study (YJ-2020-S021-01) was obtained from the Ethics Committee of the Shanghai Public Health Clinical Center, ensuring compliance with ethical standards for sample collection and antibody study. The participants involved in this study signed an informed consent form approved by the Investigational Review Board (IRB).

### Isolation and characterization of SARS-CoV-2-specific monoclonal antibodies

Peripheral blood mononuclear cells (PBMCs) from a 40-year-old COVID-19 patient, whose serum exhibited strong SARS-CoV-2 neutralizing activity, were collected on the date of discharge and stained with CD19 (BD, Cat No. 341093, Clone No. SJ25C1), IgA (Jackson ImmunoResearch Laboratories, Cat No. 109-135-011), IgD (BD, Cat No. 555778, Clone No. IA6-2.), and IgM (Jackson ImmunoResearch Laboratories, Cat No. 709-116-073) antibodies. CD19+IgA−IgD−IgM− memory B cells from COVID-19 patients were cultured in the presence of IL-21 (Invitrogen), IL-2 (Roche), and 3T3msCD40L feeder cells (NIH reagent program) in the 384-well plates as previously described[31]. After 2 weeks, the culture supernatants were collected and screened for neutralization against SARS-CoV-2. Neutralizing activity was then confirmed using RT-PCR for the immunoglobulin heavy chain (VH) and light chain (VL) genes of those wells that exhibited neutralizing activity. Finally, antibodies were expressed by HEK293F cells and purified using protein G beads (Smart-Lifesciences) for further analysis.

### Production of pseudoviruses

Pseudoviruses in this experiment were produced using a two-plasmid system. The first plasmid pcDNA3.1-spike plasmid expressed the spike gene, while the second plasmid pNL4-3.LucR⁻E⁻ carrying the information necessary for single-cycle infectious pseudoviruses. Spike genes of VOCs and Omicron variants were synthesized (GenScript) and inserted into the pcDNA3.1 vector. BA.2.75 and BA.5 subvariants plasmid were created by PCR using site-directed mutagenesis (Supplementary Table S7). Primers were designed from the Quick-Change Primer design tool (Supplementary Table S8). Spike expression plasmid and backbone plasmid were co-transfected into 70–80% confluent HEK293T cells using EZ Trans reagent (Life-iLab, China). After 28 h transfection, the pseudoviruses were harvested, aliquoted, and stored at −80 °C until use.

### Pseudovirus neutralization assay

For the neutralization assay, $1 \times 10^4$ Huh-7 cells were plated to a 96-well plate. The cells were cultured for 12–18 h at 37 °C in 5% $CO_2$. Antibodies were serially diluted and incubated with normalized pseudotyped virus for 30 mins at 37 °C. After incubation, the supernatant of Huh-7 was removed, and then a 50 µL mixture of antibody and pseudovirus was added to the 96-well plate. After infection for 24 h, 150 µL medium was supplied and cultured for another 48 h. The supernatant was removed and Huh-7 cells were lysed with 50 µL lysis buffer for 30 mins. For luciferase expression detection, 30 µL cell lysate was transferred into a new 96-well plate and 40 µL luciferase substrate (Promega) was then added. The chemiluminescence signals were collected by PerkinElmer Ensight. The neutralization activity $IC_{50}$ was calculated using GraphPad Prism9.0.

### ELISA

The RBDs of SARS-CoV-2 variants spike protein (aa Arg319-Phe541) with 8× histidine at the C-terminus was expressed by transfecting HEK 293F cells. RBDs were coated at 1 µg/mL in Carb buffer (10 mM $Na_2CO_3$, 40 mM $NaHCO_3$, pH 9.6) on a 96-well microtiter plate overnight at 4 °C. Plates were blocked with 5% non-fat powdered milk with 1% FBS in PBS. Antibodies were serially diluted in disruption buffer and transferred into a PBST washed microtiter plate. After incubation for 1 h, the plate was washed, and goat-anti-human IgG antibodies conjugated with HRP (Jackson ImmunoResearch, Code No. 109-035-098, Lot. 164726) were added (1:2500 dilution). Antibody binding was then developed using ABTS substrate solution, and absorbance was detected at 405 nm. Absorbance Absorb data were collected by BioTek Epoch 2.

### Antibody affinity by BLI

Binding affinities of antibodies with different RBD variants were performed in Octet Ni-NTA (NTA) Biosensors (Sartorius). Recombinant RBD His-tag proteins were loaded on NTA Biosensors and binding kinetics were analyzed by following gradient dilution concentration of Ab and monitoring association for 300 s. Then monitoring dissociation while the sensor was immersed in the PBST buffer for 300 s. The experiment was conducted on an Octet RED96 instrument (Sartorius) and data was analyzed using a 1:2 binding model by FortéBio Data Analysis 8.1 software.

### ACE2 competition assay by BLI

As for the ACE2 competition binding assay, 600 nM antibody was incubated with 100 nM RBD-8his protein for 30 min earlier. Procedure is similar to Ab competition assay. First, ACE2-Fc was captured by an AHC biosensor for 600 s. Second, after a baseline of 120 s in PBST buffer, the sensor was blocked with IgG1 isotype control for 600 s. Third, the sensor was soaked into the antibody-RBD pre-mixture for 600 s after baseline again in the buffer. A mixture of ACE2-Fc, IgG isotype, and RBD was as a positive control, and a mixture of isotype mAb VRC01 and RBD was as a negative control. Percent competition values were calculated based on the values of maximum binding. The binding curve of ACE2-Fc, IgG isotype, and RBD was employed as a positive control and set as a 100% competition value, while the binding curve of mAb VRC01 and RBD served as a negative control and was designated as a 0% value.

### Antibody competition assay by BLI

Antibody competition binding assays were performed on the Octet RED96 instrument as previously described[14]. Briefly, 100 nM RBD-8his protein and 600 nM second mAb were incubated for 30 mins in advance. The first mAb were captured on anti-human Fc (AHC) biosensors at 20 µg/mL for 300 s, and then the biosensor was blocked by immersing into isotype control IgG1 at 50 µg/mL. After blocking for 300 s, the biosensors were placed into a mixture of RBD-8his and second mAb to detect the first mAb-binding RBD. The specific binding signal was normalized by subtracting the self-mAb control.

### Construction expression and purification of Bispecific antibodies

Bispecific antibodies G7-Fc and 7G-Fc utilizing the formats were constructed as previously described[24]. GW01 scFv and 7F3 scFv followed the order VL-(GGGGS)3-VH. The scFvs for each antibody were synthesized by Genscript company and connected with a (GGGGS)4 linker, followed by IgG1 Fc fragment (Hinge-CH2-CH3). Thus, the sequence of G7-Fc is GW01VL-(GGGGS)3-GW01VH-(GGGGS)4-7F3VL-(GGGGS)3-7F3VH-Hinge-CH2-CH3. While 7G-Fc was designed with 7F3 scFv in the N-terminal.

HEK 293F cells (ATCC) were cultured in SMM 293-TII medium (Sino Biological Inc.) and used to express antibodies. The Ab-encoding

DNA plasmid was transiently transfected into HEK 293F cells using EZ Trans (Life-iLab, China) with a 1:3 mass-to-volume ratio. After 3 days, cells were supplemented with fresh SMM 293-TII medium. On day five post-transfection, the supernatants were collected, and the antibody was affinity purified with protein G beads. The purity and molecular weight of Ab were then future analyzed by SDS-PAGE.

### Neutralizing assay by focus reduction forming assays (FRNT)

FRNT was performed in a certified Biosafety level-3 lab, as previously described[32]. In total, 50 μL of serially diluted antibodies were incubated with 50 μL of BQ.1 or XBB authentic virus (180 focus forming unit, FFU) in 96-well microwell plates for 1 h at 37 °C. The mixtures were added to 96-well plates seeded with Vero E6 cells (ATCC, Manassas, VA) and incubated for 1 h at 37 °C. Inoculums were then removed before adding the overlay media (100 μL MEM containing 1.2% Carboxymethylcellulose, CMC). The plates were then incubated at 37 °C for 24 h. Overlays were removed, and cells were fixed with 4% paraformaldehyde solution for 30 min. Cells were permeabilized with 0.2% Triton X-100 and incubated with cross-reactive rabbit anti-SARS-CoV-N IgG (Sino Biological, Cat 40143-R001) for 1 h at room temperature (RT) before adding HRP-conjugated goat anti-rabbit IgG(H + L) antibody (1:4000 dilution) (Jackson ImmunoResearch, West Grove, PA). Cells were further incubated at RT. The reactions were developed with KPL TrueBlue Peroxidase substrates (Seracare Life Sciences Inc, Milford, MA). An EliSpot reader (Cellular Technology Ltd., Shaker Heights, OH) was used to calculate the numbers of SARS-CoV-2 foci.

### Prophylactic and therapeutic efficacy of G7-Fc against XBB.1 infection of mice

The animal study was approved by the Institutional Animal Care and Use Committees of Affiliated First Hospital of Guangzhou Medical University (Approval Number: 20230615). 6-week-old ACE2 K-18 female mice ($n = 5$/group) and BALB/c female mice ($n = 4$/group) were used to evaluate the prophylactic and therapeutic efficacy of G7-Fc in the animal study. Mice were injected either intraperitoneally (200 μg/mouse) (i.p.) or intranasally (20 μg/mouse) (i.n.) with bispecific antibody G7-Fc 24 h before or 24 h after intranasal infection with $10^5$ FFU of XBB.1. Two days after XBB.1 infection, lung samples of mice were harvested for vial titration. The amount of virus per gram of lung tissue was determined after 48 h post-infection. Mice treated with PBS were used as controls.

### Expression and purification of RBD

The coding sequence of RBDs (Arg319-Phe541) tagged with a C-terminal 8×His tag was cloned into pSecTag mammalian expression vector. The coding sequence of human ACE2(Met1-Ser740) was fused to IgG1 Fc and cloned into the pSecTag vector. The recombined RBDs and hACE2-Fc were both expressed in HEK293F cells. The expression plasmid was transiently transfected into cells. After 5 days, the medium was collected and filtered with a 0.45 μm filter. RBDs were purified by His affinity using Ni Sepharose while hACE2-Fc was purified by Protein G beads.

### Expression and purification of SARS-CoV-2 XBB spike trimer

The XBB S HexaPro ectodomain was cloned into the pcDNA3.1 vector. The expression plasmid was transiently transfected into HEK293F cells using polyethyleneimine and cultivated for 72 h. XBB S trimers were purified from supernatants by affinity column Histrap HP (GE Healthcare). The protein was further purified by gel filtration chromatography using a Superose 6 increase 10/300 column (GE Healthcare) in 20 mM Tris, pH 8.0, and 200 mM NaCl.

### Formation of XBB S/G7-Fc complex

The XBB S trimer was mixed with G7-Fc in a 1:1.3 molar ratio, incubated at 4 °C for 0.5 h, and further purified by gel filtration chromatography on a Superose 6 increase 10/300 column (GE Healthcare). The peak fraction was concentrated to 0.5 mg/ml in 20 mM Tris, pH 8.0, and 200 mM NaCl for cryo-EM study.

### Negative staining EM sample preparation and imaging

For XBB S, XBB S/G7-Fc, XBB S/G7-Fab, XBB S/7F3 IgG, XBB S/GW01 IgG, XBB S/7F3 IgG/GW01 IgG, XBB S/4L12-7F3-Fc or XBB S/REGN10989-7F3-Fc, 5 μL samples (0.02 mg/mL) were loaded to the glow-discharged carbon-coated grids (Electron Microscopy China) for 1 min. Then, the grids were blotted with filter paper and negatively stained with 2% (w/v) uranyl acetate, gently blotted, and air-dried. Images were collected with Talos L120C TEM (Thermo Fisher Scientific) using a Ceta2 camera at a nominal magnification of 73,000×.

### Cryo-EM sample preparation

Three microlitre complex was applied on a freshly glow-discharged holey amorphous nickel-titanium alloy film supported by 400-mesh gold grids, plunge frozen using the Vitrobot IV (FEI/Thermo Fisher Scientific) with a blot force of −2 and 3.0 s blot time at 100% humidity and 4 °C.

### Cryo-EM data collection and image processing

Cryo-EM data were captured on a TITAN Krios G4 TEM (Thermo Fisher Scientific) equipped with a Falcon 4i camera and a Selectris X Imaging filter (Thermo Fisher Scientific) setting to a slit width of 20 eV. Automated data collection was performed with EPU software at 300 kV in AFIS mode at a nominal magnification of 105,000×, with a physical pixel size of 1.19 Å and defocus values ranging from −1.0 μm to −3.0 μm. Each EER movie stack was dose-fractioned to 1737 frames with a total exposure dose of about 50 e−/Å².

Movie stacks were imported to Relion3.1, then motion corrected (binned 2 and dose weighted) by MotionCor2[33], and CTF estimated by Gctf[34]. All micrographs were imported to cryoSPARC[35] v4.0.3 for particle picking and 2D classification. A total of 1,650,604 good particles from 6332 selected micrographs were imported to Relion, and one round of 3D classification resulted in 3.75 Å of trimer dimer. Local refinement is performed in cryoSPARC and results in 3.07 Å with a mask around the top monomer and in 3.05 Å with a mask around the bottom monomer. Then, the particles were expanded C3 symmetry in Relion, and further local 3D classification with a mask around RBD and Fab was applied. The best class is exported into cryoSPARC, and local refinement yields a resolution of 3.0 Å.

The resolution was estimated according to gold-standard Fourier shell correlation (FSC) 0.143 criterion. All the data processing procedures are summarized in Fig. S6. The maps were sharpened by DeepEMhancer[36], and handedness was corrected using UCSF Chimera[37].

### Model building and refinement

The initial model of XBB S was generated from Omicron S-FD01(PDB: 7WOW), while G7-Fc was predicted using swiss-model[38]. Initial models were fitted into the maps using UCSF Chimera and then manually adjusted using COOT[39], and several rounds of real space refinement were performed using PHENIX[40]. The RBD domain bound with GW01 and 7F3 were refined was refined against the local refinement map of RBD-G7-Fc and then docked back into the composite map, and the whole model was refined against the composite map. Model validation was performed using Molprobity. Figures were prepared using UCSF Chimera and UCSF ChimeraX[41]. The statistics of model refinement and data collection are listed in the Supplementary Table S2.

### Reporting summary

Further information on research design is available in the Nature Portfolio Reporting Summary linked to this article.

## Data availability

Coordinates and maps associated with data reported in this manuscript were deposited to the Electron Microscopy Data Bank (EMDB) and Protein Data Bank (PDB) with accession numbers EMD-36423 and PDB 8JMM (XBB S trimer-dimer/G7-Fc), EMD-36321 and PDB 8JIN (local refined map of XBB S/G7-Fc). Accession number of GW01 heavy chain and light chain are OP480801.1 [https://www.ncbi.nlm.nih.gov/nuccore/OP480801.1/] and OP480802.1 [https://www.ncbi.nlm.nih.gov/nuccore/OP480802.1] in GenBank. Source data in this study are provided in the Supplementary Information/Source Data file. Source data are provided with this paper.

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

## Acknowledgements

We thank the Center of Cryo-Electron Microscopy, Core Facility of Shanghai Medical College, Fudan University, for the support on cryo-EM data collection. This work was supported by R&D Program of Guangzhou Laboratory (SRPG22-003 to L.S.), Shanghai Municipal Science and Technology Major Project (ZD2021CY001 to J.H.), the Ministry of Science

and Technology of China (2021YFC2302500 to L.S.), the National Natural Science Foundation of China (32370995 and 31771008 to J.H. and 3197014 to Z.C.), National Key Research and Development Program of China (2023YFE0118200 to F.W.), the "Shuguang Program" of Shanghai Education Development Foundation and Shanghai Municipal Education Commission (22SG16 to F.W.), the National Science and Technology Major Projects of China (2017ZX10202102 to J.H. and 2018ZX10301403 to F.W.), Shanghai Municipal Commission of Health and Family Planning (2018BR08 to J.H.).

## Author contributions

J.H., L.S., and F.W. conceived and designed the experiments. J.H. and F.W. performed B cell sorting and antibody cloning. Y.D.W., J.P., Y.M., P.R., J.C., and L.L. constructed the bispecific antibodies and SARS-CoV-2 pseudovirus mutants, purified antibodies, and performed neutralization assay, ELISA, and bilayer interferometry experiments. L.S., Z.C., A.H., Q.M., and Y.J.W. were responsible for the structural studies. J.Z., Z.Z., Y.Q.W., and X.X. were responsible for the authentic virus experiments. Y.D.W. and J.H. created the experiment schematic diagram. J.H., Y.D.W., Y.M., F.W., L.S., A.H., and Z.Z. analyzed the data. Y.M.W. supervised the project. J.H., L.S., F.W., Y.D.W., and A.H. wrote the paper.

## Competing interests

Patents about the G7-Fc and 7G-Fc bispecific antibodies in this study are pending. The remaining authors declare no competing interests.
