## [Peer Review File · Nature Communications]

REVIEWER COMMENTS

Reviewer #1 (Remarks to the Author):

SARS-CoV-2 has the propensity to mutate and develop resistance against current therapies. To combat this problem for Omicron variants, Wang et al. have isolated and generated mutations in an antibody called 7F3 to determine its epitope on the SARS-CoV-2 spike RBD. They then generated two bispecific antibodies using 7F3, which binds the RBM, and GW01, which binds outside the RBM. They did further characterizations using one of the bispecifics, including binding studies to RBDs of different variants, neutralization assays using pseudoviruses, in vivo studies in mice, and determined its structure in complex with the spike using cryo-EM. They also proposed a neutralization mechanism based on additional binding studies.

The bispecific antibody, G7-Fc was able to neutralize the Omicron variants, likely due to its ability to bind to a highly conserved region of the RBM. The bispecific was also effective in mice when used both prophylactically and therapeutically. Most noteworthy is the cryo-EM structure determined in complex with the spike and bispecific antibody which revealed the crosslinking of two trimers. The data presented would be of interest to others in the field since the virus is evolving and new treatments would be useful. However, there are still some questions that remain regarding the interpretation of several of the results as indicated below. Some figures also need to be updated to help with interpretation.

Major comments:

1. What was the purpose of the glutaraldehyde cross-linking? Was it only used for the gel or was it used for the bispecific antibody preparation? If the latter, has any NS-EM and/or cryo-EM been done in the absence of glutaraldehyde cross-linking?

2. Supplementary Figure S1 – Does the gel filtration curve indicate that a complex forms with two trimers? Including molecular weight standards would be helpful in interpreting the chromatogram. This would also help determine whether or not the crosslinking is an artifact from grid preparation somehow.

3. How is it that the Fc regions crosslink two trimers?

4. While the bispecific antibody worked prophylactically in mice, how would this translate to the human population? Does that antibody have to be administered a certain timeframe before infection? Some speculation on this would be helpful.

Other comments:

Figures:

5. A schematic, like that of the bispecific antibody in Figure 2A would be helpful to have to show how the crosslinking with S trimers occurs. It's not quite clear from Figure 4 due to the color scheme (see below).

6. Figure 4 – It would help to label the antibodies in the side views of panel A and possibly consider a different color scheme. The color of GW01 and stem region of the spike look similar to me. It would also help to make the color scheme consistent between panels A, B and C.

7. Figure 5 – it would help to have the different CDRs colored differently in panel A. It's difficult to tell with just the labeling where each CDR loop is. Distances should be provided in panels B and C.

Methods –

8. What concentration of complex was used for preparing grids? Was it 0.5mg/ml? Also, what buffer was used?

Minor points:

Line 35 – assay is spelled incorrectly

Line 102 – I think “region” would read better here rather than “proteins”

Line 139 – finding should be plural

Line 158, 178 – antibodies should be singular

Lines 251-252 – this is an incomplete sentence

Line 263 – It should be mentioned that VRC01 is an HIV antibody. The general reader won't necessarily know this bnAb.

Line 304 – incorporating is misspelled

Reviewer #2 (Remarks to the Author):

The manuscript by Wang et al. describes the discovery and testing of a human monoclonal antibody with extraordinary breadth against SARS-CoV-2, including Omicron VOCs and sarbecoviruses.

This highly collaborative set of investigators has an impressive track record for antibody discovery, and this manuscript demonstrates their continued groundbreaking work. The paper described in vitro characterization, in vivo testing, and structural analyses that provide important information as to why this new monoclonal, and its bispecific derivative, are so much more potent and broad compared to other RBD-binders.

This groundbreaking work has high significance for the field, in that not only is there increased knowledge about immune targeting of highly conserved regions in SARS-CoV-2 (not seen previously), but the authors also describe an exciting bispecific entity that could provide therapeutic or

prophylactic benefit as the world moves into the beginning of the 4th year of SARS-CoV-2 outbreaks and spread.

Overall the methodology is sound. The in vitro data are sound (neutralization binding, biochemistry), accompanied by mapping of residues in RBD to indicate that the new antibody 7F3 is not as sensitive to changes as antibodies discovered earlier. The bispecific constructs show that combining 7F3 with GW01 with the 7F3 epitope on the N-terminus is far more effective. Further work to understand how the bispecific interacts with RBD includes cryoEM studies that help to show the novelty of 7F3 in combination with the noncompeting GW01 combining site.

My major concern is that the animal model data, while certainly encouraging, may not portend as much success in nonhuman primates or humans when tested there. The data show that there is on the order of a 1 log₁₀ reduction (to the limit of detection) in focus forming units from a peak of 1,000 in the controls animals. In other models, there can be much more virus replication in the lung that is at least 2 orders of magnitude greater than what is reported for these mouse data. Thus the authors should be more cautious in their statements about prophylactic and therapeutic benefit. Certainly, the data shown are encouraging. But they need to be followed up with other experiments to determine whether the bispecific is more useful when given prophylactically or therapeutically, in models that have more pathogenesis. I do not see that these experiments are needed for this manuscript, rather that the results and discussion should be modulated to state that they have achieved a 1 log₁₀ reduction and more work is needed.

The work holds great promise for a novel therapeutic agent which may be effective when given by the aerosol route. Overall, this is a complete and important piece of work that significantly advances the field.

Reviewer #3 (Remarks to the Author):

Summary:

Wang and colleagues report the isolation of antibody 7F3 that binds to RBD and neutralizes SARS-CoV-2 variants XBB, XBB.1.5, XBB.1.16 and XBB.1.16.1. Next, they combine this antibody with a previously isolated antibody, GW01, into a bispecific antibody. GW01 has poor to no neutralizing activity against Omicron lineage, BQ.1.1 and XBB variants but has sub-microgram/mL neutralizing activity against SARS-CoV (i.e. SARS1), RS3367 and WIV1, which are SARS-CoV like sarbecoviruses.

Two bispecific antibodies were made, each arm containing one 7F3 and one GW01 per arm: (1) G7-Fc bispecific having GW01 furthest away from the Fc domain and (2) 7G-Fc bispecific having 7F3 furthest away from the Fc domain. Of these two, 7G-Fc did not retain activity against all the variants—thus, they continued studies with G7-Fc (G7). Next, they test G7's protection against virus growth in the lungs of wild type BALB/C mice by G7 when given intraperitoneally or intranasally 24h before or after inoculation of XBB.1 virus. They show that lung virus titers were below the limit of detection in treated animals (~2.2 logs) vs ~3.2 logs in PBS treated animals. Next, they use Cryo-EM analysis of G7 binding to XBB spike trimers and show how 7F3 and GW01 binding occurs in the context of the bispecific. This shows G7 binding with RBD up (or open) and that there are extensive hydrogen bonding interactions are major contributors to the modes of recognition of binding. They also use structural information to identify 4 mutations in the XBB that leads to decreased neutralization activity for G7-Fc.

This paper seeks to provide a mechanistic look at a new antibody, 7F3, and how it functions in the context of a bispecific antibody. However, there are limitations to the study/claims that need to be significantly revised and/or supported by additional experiments. In addition, there are methodologic details and ethical statements that are missing.

Concerns:

1) Line 88 and elsewhere, At times the manuscript appears to claim that G7-Fc shows the broadest neutralization of variants and of sarbecoviruses. If this is indeed the claim, it is a rapidly moving target and very subjective claim (i.e., how is breadth defined in either case). It should therefore be removed or modified to make the claim less subjective. Similarly, the statement line 131-132 about the uniqueness for neutralizing the combination of BQ.1.1, XBB.1.5 and XBB.1.16 should be removed.

2) Lines 98-100 and Methods. Almost no information is provided on the isolation of the antibody 7F3 in the manuscript text and no method is provided. Furthermore, there is no ethics statement or clinical trial information relating to the sample from which 7F3 was isolated from. This information should be provided. The method should include information about what virus the subject was infected with (or month/year), how the PBMCs were sorted, what probe was used, and how the antibody was sequenced. There should be a supplemental figure showing the flow sorting scheme. There is also no method on the 7F3 cloning, expression and purification.

3) Throughout the manuscript ng/mL or ug/mL are used. However, since the bispecifics are a different average molecular weight than the antibodies they are being compared to, it is more appropriate to express the data in molar values. This will allow a more accurate comparison of the monospecific and bispecific antibodies binding and neutralization activity. Please convert all data to molar.

4) Line 114-116 says "LY-CoV1404 exhibited high neutralization potency against various sarbecovirus strains except for the Omicron subvariants ...". Please revise because LY-CoV1404 does not neutralize any of the SARS-CoV (SARS1) like sarbecoviruses. It only neutralizes SARS-CoV-2 like sarbecoviruses.

5) Paragraph for lines 110-122. While most readers should be familiar with the concept of SARS-CoV-2 variants and SARS-CoV (SARS1) from the scientific and lay media, most are not going to be familiar with other sarbecoviruses. Therefore, the manuscript would be improved by introducing to the names of the viruses that are being tested, the rationale for testing them and that they are SARS-CoV (SARS1) related viruses in the sarbecovirus family. Prior to showing SARS-CoV (i.e. SARS1), RS3367 and WIV1 data. Formally, the non-SARS-CoV-2 viruses are Clade 1a sarbecoviruses and SARS-CoV-2 variants are clade 1b sarbecovirus.

6) Lines 117-120. Please provide information in the manuscript on what the specific test for synergy was performed for the data referred to in figure 1d. Was it a formal synergy calculation? If so, please provide the details in the text and method section. If it was simply the combination experiment without a formal synergy calculation they cannot claim a test of synergy. That said, one can occasionally make a claim of potential synergy when one notes a large (i.e., >10-fold) improvement in neutralization over that of the most potent parental antibody in a combination. This sentence and elsewhere about synergy in the discussion should be revised appropriately.

7) Section starting at lines 123. Binding modes for RBD targeting antibodies are generally discussed in the literature in terms of the Barnes classification schema. This schema classifies antibodies by the position of RBD required for binding (i.e., Up or both Up and Down; also known as open or open/closed, respectively) and whether or not their epitope includes the ACE2 binding site (i.e. RBM). It should be noted that Class III antibodies do not bind in the RBM but can sterically block ACE2 binding. The manuscript would be greatly improved by discussing the binding of 7F3 and GW01 in terms of this classification schema and then comparing the binding modes to other similarly classified antibodies. This will enable the reviewers and readers to better interpret the data and their significance. For example it appears from the graphics that 7F3 may be a Class I and GW01 may be class III but this reviewer is not clear without more information from the authors. If that is the case that the better comparison antibodies for 7F3 would be another class I mAbs that has a similar epitope and for GW01, LY-CoV1404 and S309 would be good comparators.

8) Line 129-131. This sentence interprets the data in Figure 1f as meaning that 7F3 binds within the RBM (i.e., ACE2 binding site). However, an alternate hypothesis that one can draw from the data in 1f is that 7F3 might sterically block ACE2 from binding to the RBM. Structural analysis is necessary to distinguish between these two possibilities. This is done later in the paper and can be concluded only after the structural data is shown/discussed. Please modify the manuscript appropriately.

9) Line 138-141. This states that, "These findings suggest that 7F3 epitope likely encompasses multiple amino acid residues within the RBD, given its retention of binding to all single and triple mutants tested. 7F3 targets a conserved, functionally critical epitope within the RBM that distinguishes it from other characterized antibodies."

a. In lines 138-140, it is not clear what the significance of encompassing multiple amino acids is because by definition all antibody epitopes encompass multiple amino acid residues. Please revise and clarify what is intended.

b. It is not necessarily true that the epitope needs to be conserved or functionally critical. For example, the paratope of the antibody may simply accommodate side the new side chains from mutated residues while maintaining contacts to the RBD. Similarly, just because a residue has not

mutated yet, does not mean that it is critical. It may simply be that we have not yet observed it yet. This is illustrated by the continued appearance of mutations at different locations in the Spike of SARS-CoV-2 and the dozens of new mutations found in the recently identified BA.2.86. Mutations may also impact the relative open or closed (i.e., up/down) state of spike or create a conformational change in RBD that modify the structural binding area of 7F3. These changes might occur outside of the binding site of 7F3 and therefore be indirect.

c. To support the conclusion the mutations are in a functionally critical region of the spike RBD or even the RBM, they would not only need to show data that the mutations significantly decrease ACE2 affinity for spike but also that viral titers are significantly decreased relative to viruses containing the parental spike proteins.

10) Line 162 states that 7G-Fc is “less efficient”. This is subjective. Is it less potent neut, binding both, something else (i.e., kinetics, neut breadth)? Please be specific so a judgement can be made on the validity of the statement.

11) Section on beginning on Line 182 on the prophylactic and therapeutic efficacy of G7-Fc. The BALB/C model is not the model of choice for testing antibody therapies for SARS-CoV-2. The preferred model is the Syrian golden hamster model. In addition, even amongst mouse models, it is not the best choice as it uses the mouse ACE2 receptor. When a mouse model is used, the best is the human ACE2 mouse knock-in model. Finally, there are only 4 animals in each group (standard is 10) and there is only a 1 log difference in peak viral lung titer and the level of detection limit. This does not provide a robust measure of this antibody’s potential efficacy. Taken together this severely impacts the significance of the animal experimental findings.

12) Section beginning on Line 197 on the Cryo-EM structure

a. The binding should be put in the context of the Barnes classification and compared use to the functional and biophysical experiments previously shown in the paper. Does the structure support and/or explain the results? That is please explain how the structure supports the proposed mechanisms of action. How does binding compare to other similarly classified RBD targeting mAbs. This is done to some extent for LY-CoV1404 but this is a better comparator for GW01 not 7F3. Also, S309 does S309 show more similarity to GW01 than LY-CoV1404?

b. Based upon the neutralization data and affinity data in Figure 1, one would expect that the GW01 variable fragment (Fv) would not bind to the XBB spike protein. However, the manuscript shows that the GW01 Fv domain is binding. This is surprising. There is not a convincing argument as to why this is observed. If it was simply a loss of binding surface area or a mutation that impacted binding affinity that leads to a loss or decrease in neutralization one should still expect to see binding in a kinetics assay in figure 1b/c, but they do not detect binding to XBB in those panels. This is not adequately explained.

c. Line 207-210. Please clarify in the sentence whether the binding to RBD by 7F3 and GW01 domains are from the same G7-Fc molecule, and if it is from the same molecule is it from the same or different arms. Similarly, I believe what line 209-210 is stating is that one arm of an antibody binds one spike and the other arm of the same antibody binds the other but the wording states it is the Fc domain that does the crosslinking. This is awkward and not formally correct since the Fc domain

does not bind the spike. Please provide several cartoon viewpoints in the main figure or a supplemental figure showing the binding mode observed in the CryoEM and how the binding is occurring. This is important to interpreting what is happening.

d. Are there structures of parental antibodies alone and/or as a mixture? If so how does their binding compare to what is seen in the CryoEM? If there is not, at minimum, please provide negative stain EM reconstructions or low-resolution Cryo-EM to determine if the bispecific induces binding modes that are different than parental antibodies do by themselves. This is especially important in light of the finding that the GW01 Fv domain is binding spike. For example, is the binding of 7F3 inducing a conformational change in a mixture or bispecific format that then allows GW01 to bind?

e. Line 207. Since each of the three RBDs on spike naturally and independently switch between up and down states it is not formally possible to know from the provided data whether binding of G7-Fc induces a 3-RBD up state or if it simply when RBD goes into an up position G7-Fc lock it in the up state while it remains bound. In the Cryo-EM experiment G7-Fc is in excess so it is more likely to capture in a steady state saturated binding with RBD bound than unbound. This data indicates that G7-Fc can bind up to 3 RBDs in the up state.

f. It is very interesting that there are a large number of hydrogen bond networks being created between RBD and 7F3. This is more than is typically seen and should be explored and compared to other antibodies that bind the same region of RBD. What about salt bridges? How were the surface area and hydrogen bond contacts determined? Was PISA used? If so, please provide the PISA data as a supplemental table/figure and add it to the methods.

g. Lines 228-231 describes a reduction in binding surface area for GW01 of 56.6 angstroms in XBB spike relative to BA.1 Spike. There is no reference provided. Also, what was the overall buried surface area in both cases? Given that there was likely >500 angstroms of buried surface area remaining on XBB, it seems unlikely by itself to explain the total loss of binding and neut for XBB in Figure 1b/c/d. Were other binding modes explored (i.e, salt bridges, charge interactions)?

h. Line 243-245 describes the frequency of the Y501A mutation. There is no reference for the database used and searched in the methods and text. The text should also state which database was used. In addition, while the frequency of 501 alanine may be low there are significant numbers of changes at position 501 other than alanine. They should also note the other amino acids changes that are in the database at each position in the database they search. The burden is on the authors to show that the other mutations do not impact activity. In the absence of data showing the other mutations do not impact binding and neut the assumption should be that ANY change at the residues noted in Figure 5d will disrupt G7-Fc activity.

i. Line 256-258 describes a G7-Fc epitope. Since this is a bispecific antibody it is best to refer to the individual epitopes of each Fv domain independently rather than including them as one large epitope. This is because mutations in each component Fv epitope should have independent effects on the molecule. Therefore, when discussing mutations and their effects it is best to put them in the context of the Fv domain that is being impacted based on the structure.

13) Line 267-270. What data is being used to support the statement that G7-Fc synergistically induces the formation of trimer dimers? It is unclear to this reviewer what data in the manuscript supports

this conclusion. I do not believe the data in figure 4 supports this conclusion. Also, there is no evidence of synergistic neutralization. Please modify the manuscript to provide a clarification of how the data supports the conclusion or remove the statement.

14) Line 295-299. What data supports the finding that the binding of G7-Fc induces conformational change that allows GW01 to bind. I did not see experiments showing this in the manuscript. High-resolution structures of the unbound spike, spike with each parental mAb alone and spike with mixtures of the two mAbs. Both XBB and a spike that is known to bind both GW01 and 7F3 are needed to be compared and identify a conformational change. In addition, an order of addition experiment using BLI that shows that the binding of 7F3 first allows GW01 to bind is needed. Without these data, the claim should be removed.

15) Figure 1d: Please indicate what the ratio of 7F3 and GW01 were in the combination neutralization assay.

16) Figure 1e/f. It is not clear what the order of addition is for the figures. Also, please provide a percent competition value as a table in this figure or in the supplement that clearly indicates what the competitor is and what the analyte is. One would expect ACE2 self-competition to be near 100% it appears to be <50%, which is very weak and would indicate that this experiment is not interpretable. Please provide the table and if self-competition is < 75% it needs to be repeated.

17) Figure 2f: Because there are more virus variants tested in this table than in figure 1d, this data table should include a comparison to the parental Abs and the mix of the parental antibodies as controls.

Other comments:

1) Line 81-82: This sentence says the rapid spread of SARS-CoV-2 variants make it urgent to develop pan-sarbecovirus antibodies. However, it is not clear to this reviewer how the appearance of SARS-CoV-2 variants means we need to have better SARS-CoV (SARS1) like antibodies. I recommend adding the logic or modifying the sentence.

2) Line 104. The generic name bebtelovimab should be mentioned with the first use of LY-CoV1404 earlier in the paper on line 66. Similarly, earlier Evusheld should have the generic names provided.

3) Line 106. The manuscript would be improved by providing a few more details about GW01 such as where it binds, is it a human antibody. It says it is a sarbecovirus Nab. So a reader may want to know if it was isolated to target SARS-CoV (SARS1) and then found to have SARS-CoV-2 activity or was it isolated against SARS-CoV-2 and found to have SARS1 breadth?

4) Line 107-109, indicates the affinities in Fig 1b and 1c accord with ELISA results from 1a. Since multiple approaches can be used to determine binding (and affinity), a reader may appreciate mentioning in lines 101-103 that the binding there is ELISA and then say that the affinities were measured using biolayer interferometry in lines 107-109.

5) Section on construction of bispecific antibodies starting at line 144. Since there are many formats and types of bispecific antibodies, the manuscript would be improved by a short description of the

bispecific antibody format being used. Of particular note, this is a tetravalent bispecific. Does it contain Fc mutations (i.e., knob and hole mutations, Fc-effector function mutations, etc)? What is the order of the variable fragments on each arm in each version of the molecule. For this last item, a graphic showing both versions being made and tested should be provided in the main figure or a supplemental figure.

6) Line 164 states the order increases the binding affinity. This is not formally correct. The order impacts the binding affinity. Please change the wording to either say having X first increases the affinity or that the order impacts the affinity. Also, they may want to discuss the binding relative to the neutralizing activity.

7) Line 170-172. Why is it surprising that G7-Fc neutralizes the viruses panel shown? Based on the parental antibody neutralization and combination experiment, one would expect it to neutralize as it does. So it would be surprising if it did not. Please rephrase.

8) Line 343. Since the molecule being tested are not Fabs but mAbs, which are bivalent, or bispecific, which are tetravalent, a 1:1 binding model are typically not the correct binding model to use for kinetic (affinity) analyses. For a standard mAb with an Fc domain a bivalent binding models is most appropriate to account for inherent avidity. For a tetravalent molecule it is not clear what model is best to use but a bivalent model appears more appropriate than a 1:1 binding model.

9) Competition assay by BLI method. This method does not explain the ACE2 binding competition assays. This is critical because it is not clear what order the assay is being performed in figure 1 and 4.

10) The graphs in figure 1b and 2d are too small to see. I suggest that all or only a select subset be included in the figure and the remaining be placed into a supplement to allow better visualization of the graphs.

11) Figure 1d and 1f. The dark fonts in red/green colored boxes are difficult to read. Suggest changing to a light-colored font in dark boxes. In addition, figure 1d is not color blind friendly with green next to red boxes.

12) The legend description for panel (f) is shown as (e)

13) Panel 2g, y-axis is labeled Inhibition rate. It is not a rate. It should be labeled as percent inhibition or neutralization percent depending on how the calculation was performed.

14) Figure 3b. To provide better identification of the measured values relative to one another, please provide minor tick marks for the y-axis and provided the log value for the lower limit of detection in the legend.

15) It is difficult to figure out what is 7F3, GW01 and RBD in the models of the upper left-hand panel of figure 1a . Please label each.

16) It is difficult to see the yellow outline of the ACE2 binding site in panel 1C, please make more prominent and consider changing the color to make it more visible.

17) The manuscript switches between up and open to describe RBD's state. My preference is to cue the reader that open equals up and closed equals down. And then to stick with one or the other nomenclature for the remaining part of the article.

REVIEWER COMMENTS

Reviewer #1 (Remarks to the Author):

SARS-CoV-2 has the propensity to mutate and develop resistance against current therapies. To combat this problem for Omicron variants, Wang et al. have isolated and generated mutations in an antibody called 7F3 to determine its epitope on the SARS-CoV-2 spike RBD. They then generated two bispecific antibodies using 7F3, which binds the RBM, and GW01, which binds outside the RBM. They did further characterizations using one of the bispecifics, including binding studies to RBDs of different variants, neutralization assays using pseudoviruses, in vivo studies in mice, and determined its structure in complex with the spike using cryo-EM. They also proposed a neutralization mechanism based on additional binding studies.

The bispecific antibody, G7-Fc was able to neutralize the Omicron variants, likely due to its ability to bind to a highly conserved region of the RBM. The bispecific was also effective in mice when used both prophylactically and therapeutically. Most noteworthy is the cryo-EM structure determined in complex with the spike and bispecific antibody which revealed the crosslinking of two trimers. The data presented would be of interest to others in the field since the virus is evolving and new treatments would be useful. However, there are still some questions that remain regarding the interpretation of several of the results as indicated below. Some figures also need to be updated to help with interpretation.

We appreciate the positive comments.

Major comments:

1. What was the purpose of the glutaraldehyde cross-linking? Was it only used for the gel or was it used for the bispecific antibody preparation? If the latter, has any NS-EM and/or cryo-EM been done in the absence of glutaraldehyde cross-linking?

The bispecific antibody was designed to form a dimer through the hinge region. However, the disulfide bond of the bispecific antibody was disrupted by the SDS-PAGE gel, and we only detected the single-chain format with a size of 85kDa (Fig. 2B, right). To remedy this, we used glutaraldehyde to cross-link the bispecific antibody. After glutaraldehyde-mediated crosslinking, we found the full size of the bispecific antibody to be approximately 170kDa by SDS-PAGE (Fig. 2B, left), proving the natural dimer structure. Glutaraldehyde crosslinking was only used for the gel, while NS-EM and cryo-EM were performed without crosslinking.

2. Supplementary Figure S1 – Does the gel filtration curve indicate that a complex forms with two trimers? Including molecular weight standards would be helpful in

interpreting the chromatogram. This would also help determine whether or not the crosslinking is an artifact from grid preparation somehow.

Thank you for this suggestion. Molecular weight standards and the gel filtration curve of XBB S have been added to the chromatogram in Supplementary Fig. S4a, indicating that the complex of XBB S and G7-Fc are obviously larger than XBB S alone. In addition, the negative-staining EM images of the S trimer and S in complex with different antibodies showed that only S/ G7-Fc form di-trimers (Fig. S4c). As described above, glutaraldehyde cross-linking was only used for the SDS-PAGE gel, but not for complex purification and EM study.

Supplementary Fig. S4. Purification of SARS-CoV-2 XBB S in complex with G7-Fc. (a) Gel-filtration curve and SDS-PAGE of SARS-CoV-2 XBB S complexed with G7-Fc complex. For comparison, the gel-filtration curves of SARS-CoV-2 XBB S and the molecular weight standards were also shown. (b) SDS-PAGE of G7-Fc and G7-Fab. G7-Fab was obtained by digesting G7-Fc using papain. (c) Negative staining EM images of SARS-CoV-2 XBB S alone, XBB S/G7-Fc complex, XBB S/G7-Fab, XBB S/7F3 IgG, XBB S/GW01 IgG and XBB S/7F3 IgG/GW01 IgG, showing that only G7-Fc binding induces the formation of trimer dimer.

3. How is it that the Fc regions crosslink two trimers?

In order to test the function of the Fc region, we removed the Fc region by papain digestion. We discovered that when the Fc region was truncated, G7-Fc fab failed to induce the formation of trimer dimers (Supplementary Fig. S4b and 4c, *see the above figure*). We also added a schematic diagram to show how Fc regions crosslink S trimers in Supplementary Fig. S7b and 7c (*see below*). The Fc region is missing in the final model because of its flexibility, though weak density is observed in the 2D classification (Supplementary Fig. S5b, *see below*). The Fc regions play an important role in crosslinking two XBB trimers.

Supplementary Fig. S7. The XBB S/G7-Fc complex formation. (b) Schematic diagram of G7-Fc. (c) Schematic diagram showing how Fc regions crosslink S trimers. The red dashed line represents a G7-Fc antibody. The zoomed-in view of RBD/G7-Fc is shown on the right panel. Due to the distance limitation of (GGGGS)₄ linker (~70Å) between GW01 and 7F3, 7F3 and GW01' bound to the same RBD were derived from two G7-Fc antibodies. Yellow spheres represent the N-terminal of 7F3. Magenta spheres represent the C-terminal of GW01/GW01'.

Supplementary Fig. S5. (a) 2D classification results of G7-Fc bound XBB S.

4. While the bispecific antibody worked prophylactically in mice, how would this translate to the human population? Does that antibody have to be administered a certain timeframe before infection? Some speculation on this would be helpful.

For prophylactic applications of the bispecific antibody in human populations, the feasibility of administering the antibody within 24 hours of XBB infection is limited, given the absence of early symptoms. In a noteworthy development, Professor Jinghe Huang and her colleagues, who contracted COVID-19 in late August, employed nasal spray delivery of the G-7-Fc bispecific antibody. The rapid alleviation of symptoms such as nasal congestion, runny nose, and nasal discharge within half an hour was observed. Furthermore, the bispecific antibody induced a conversion from COVID-19 antigen positivity to negativity within two days. These compelling results from human subjects suggest a prophylactic potential for the bispecific antibody in the context of human populations.

Other comments:

Figures:

5. A schematic, like that of the bispecific antibody in Figure 2A would be helpful to have to show how the crosslinking with S trimers occurs. It's not quite clear from Figure 4 due to the color scheme (see below).

Thank you. We added a schematic diagram to show how the bispecific antibody crosslink S trimers in Supplementary Fig. S7b and 7c (*also see the response to Comment 3*).

6. Figure 4 – It would help to label the antibodies in the side views of panel A and possibly consider a different color scheme. The color of GW01 and stem region of the spike look similar to me. It would also help to make the color scheme consistent between panels A, B and C.

Thanks. We changed the color scheme of Fig. 4a-4c as suggested.

7. Figure 5 – it would help to have the different CDRs colored differently in panel A. It's difficult to tell with just the labeling where each CDR loop is. Distances should be provided in panels B and C.

Thanks for this suggestion. We changed the colors of different CDRs to make them more visible.

Methods –

8. What concentration of complex was used for preparing grids? Was it 0.5mg/ml? Also, what buffer was used?

Thanks for pointing this out. The complex fraction of Superose 6 was concentrated to 0.5 mg/ml in 20 mM Tris, pH 8.0, and 200 mM NaCl for preparing grids.

Minor points:

Line 35 – assay is spelled incorrectly

Thanks for pointing it out. It was corrected.

Line 102 – I think “region” would read better here rather than “proteins”
It was corrected.

Line 139 – finding should be plural

It was corrected as suggested.

Line 158, 178 – antibodies should be singular

It was corrected.

Lines 251-252 – this is an incomplete sentence

Thanks for pointing it out. It was corrected.

Line 263 – It should be mentioned that VRC01 is an HIV antibody. The general reader won't necessarily know this bnAb.

Thanks for pointing it out. It was corrected.

Line 304 – incorporating is misspelled

It was corrected as suggested.

Reviewer #2 (Remarks to the Author):

The manuscript by Wang et al. describes the discovery and testing of a human monoclonal antibody with extraordinary breadth against SARS-CoV-2, including Omicron VOCs and sarbecoviruses.

This highly collaborative set of investigators has an impressive track record for antibody discovery, and this manuscript demonstrates their continued groundbreaking work. The paper described in vitro characterization, in vivo testing, and structural analyses that provide important information as to why this new monoclonal, and its bispecific derivative, are so much more potent and broad compared to other RBD-binders.

This groundbreaking work has high significance for the field, in that not only is there increased knowledge about immune targeting of highly conserved regions in SARS-CoV-2 (not seen previously), but the authors also describe an exciting bispecific entity that could provide therapeutic or prophylactic benefit as the world moves into the beginning of the 4th year of SARS-CoV-2 outbreaks and spread.

Overall the methodology is sound. The in vitro data are sound (neutralization binding, biochemistry), accompanied by mapping of residues in RBD to indicate that the new antibody 7F3 is not as sensitive to changes as antibodies discovered earlier. The bispecific constructs show that combining 7F3 with GW01 with the 7F3 epitope on the N-terminus is far more effective. Further work to understand how the bispecific interacts with RBD includes cryoEM studies that help to show the novelty of 7F3 in combination with the noncompeting GW01 combining site.

My major concern is that the animal model data, while certainly encouraging, may not portend as much success in nonhuman primates or humans when tested there. The data show that there is on the order of a 1 log₁₀ reduction (to the limit of detection) in focus forming units from a peak of 1,000 in the controls animals. In other models, there can be much more virus replication in the lung that is at least 2 orders of magnitude greater than what is reported for these mouse data. Thus the authors should be more cautious in their statements about prophylactic and therapeutic benefit. Certainly, the data shown are encouraging. But they need to be followed up with other experiments to determine whether the bispecific is more useful when given prophylactically or therapeutically, in models that have more pathogenesis. I do not see that these experiments are needed for this manuscript, rather that the results and discussion should be modulated to state that they have achieved a 1 log₁₀ reduction and more work is needed.

Thank you for your valuable suggestion. As previously observed, SARS-CoV-2 Omicron and its sub-lineages have demonstrated the ability to naturally infect wild-type BALB/c mice, resulting in moderate viral replication and lung pathology. This provides a rapid and cost-effective in-vivo model for evaluating the effectiveness of anti-SARS-CoV-2 agents. Importantly, authentic viral replication in the lungs of BALB/c mice infected with Omicron variants can be detected for up to 3 days post-infection. To further assess the in-vivo impact of G7-Fc, we conducted additional experiments involving the prophylactic and therapeutic administration of G7-Fc against SARS-CoV-2 XBB.1 infection in K18-hACE2 mice. As depicted in Figure 3b (See below), G7-Fc exhibited significant protection against XBB.1 infection, with almost imperceptible levels of authentic virus present on day 2 post-infection.

Fig. 3. In vivo prophylactic and therapeutic activity of the bispecific antibody G7-Fc via different administration routes. (a) Schematic diagrams of G7-Fc in the prevention (abbreviation P) and treatment (abbreviation T) of XBB.1 infection. G7-Fc was administrated intraperitoneally (abbreviation i.p.) or intranasally (abbreviation i.n.). (b) Viral titers (FFU, focus-forming units) of lung tissue of (b) K18-ACE2 mice (n=5) and (c) BALB/c mice (n=4) were determined after 48 h post-infection. Each data point represents an individual mouse within the respective groups. Statistical significance was analyzed by t-test using Prism software (version 9, GraphPad Software) and $p < 0.001$ is expressed as ***.

The work holds great promise for a novel therapeutic agent which may be effective when given by the aerosol route. Overall, this is a complete and important piece of work that significantly advances the field.

We thank the reviewer for the above positive comments.

Reviewer #3 (Remarks to the Author):

Summary:

Wang and colleagues report the isolation of antibody 7F3 that binds to RBD and neutralizes SARS-CoV-2 variants XBB, XBB.1.5, XBB.1.16 and XBB.1.16.1. Next, they combine this antibody with a previously isolated antibody, GW01, into a bispecific antibody. GW01 has poor to no neutralizing activity against Omicron lineage, BQ.1.1 and XBB variants but has sub-microgram/mL neutralizing activity against SARS-CoV (i.e. SARS1), RS3367 and WIV1, which are SARS-CoV like sarbecoviruses. Two bispecific antibodies were made, each arm containing one 7F3 and one GW01 per arm: (1) G7-Fc bispecific having GW01 furthest away from the Fc domain and (2) 7G-Fc bispecific having 7F3 furthest away from the Fc domain. Of these two, 7G-Fc did not retain activity against all the variants—thus, they continued studies with G7-Fc (G7). Next, they test G7's protection against virus growth in the lungs of wild type BALB/C mice by G7 when given intraperitoneally or intranasally 24h before or after inoculation of XBB.1 virus. They show that lung virus titers were below the limit of detection in treated animals (~2.2 logs) vs ~3.2 logs in PBS treated animals. Next, they use Cryo-EM analysis of G7 binding to XBB spike trimers and show how 7F3 and GW01 binding occurs in the context of the bispecific. This shows G7 binding with RBD up (or open) and that there are extensive hydrogen bonding interactions are major contributors to the modes of recognition of binding. They also use structural information to identify 4 mutations in the XBB that leads to decreased neutralization activity for G7-Fc.

This paper seeks to provide a mechanistic look at a new antibody, 7F3, and how it functions in the context of a bispecific antibody. However, there are limitations to the study/claims that need to be significantly revised and/or supported by additional experiments. In addition, there are methodologic details and ethic statements that are missing.

Concerns:

1) Line 88 and elsewhere, At times the manuscript appears to claim that G7-Fc shows the broadest neutralization of variants and of sarbecoviruses. If this is indeed the claim, it is a rapidly moving target and very subjective claim (i.e., how is breadth defined in either case). It should therefore be removed or modified to make the claim less subjective. Similarly, the statement line 131-132 about the uniqueness for neutralizing the combination of BQ.1.1, XBB.1.5 and XBB.1.16 should be removed.

Thanks for your suggestion. We have subsequently revised or adjusted the statements in line 88 and lines 131-132 to mitigate subjectivity and enhance the objectivity of the claims. We revised the “broadest” to “broad” and removed the “first” in lines 131-132.

2) Lines 98-100 and Methods. Almost no information is provided on the isolation of the antibody 7F3 in the manuscript text and no method is provided. Furthermore, there is no ethics statement or clinical trial information relating to the sample from which 7F3 was isolated from. This information should be provided. The method should include information about what virus the subject was infected with (or month/year), how the PBMCs were sorted, what probe was used, and how the antibody was sequenced. There should be a supplemental figure showing the flow sorting scheme. There is also no method on the 7F3 cloning, expression and purification.

In the methodology section, we incorporated an ethics statement along with a detailed description of the B cell sorting, culture methodology, and antibody cloning processes. Peripheral blood mononuclear cells (PBMCs) were obtained from a 40-year-old male individual recovering from COVID-19, whose serum demonstrated robust neutralizing activity against SARS-CoV-2 at the time of discharge. The sorting procedure is elucidated in the supplementary material, specifically presented in Fig. S1.

Supplementary Fig. S1. The gating strategy for the isolation of the memory B cell subset (CD19+IgA-IgD-IgM-) from peripheral blood mononuclear cells (PBMC) involved a multi-step process using flow cytometry. The following steps were employed: Initially, lymphocytes were identified and gated from the overall PBMC. Within the lymphocytes gate, single memory B cells were further identified and gated. Within the single memory B cell gate, further gating was applied to exclude B cells

expressing IgA, IgD, and IgM. The IgG-positive memory B cells were selectively chosen by excluding the other Ig isotypes.

3) Throughout the manuscript ng/mL or ug/mL are used. However, since the bispecifics are a different average molecular weight than the antibodies they are being compared to, it is more appropriate to express the data in molar values. This will allow a more accurate comparison of the monospecific and bispecific antibodies binding and neutralization activity. Please convert all data to molar.

Thank you for pointing this out. We converted all the data in molar values for a more accurate comparison.

4) Line 114-116 says “LY-CoV1404 exhibited high neutralization potency against various sarbecovirus strains except for the Omicron subvariants ...”. Please revise because LY-CoV1404 does not neutralize any of the SARS-CoV (SARS1) like sarbecoviruses. It only neutralizes SARS-CoV-2 like sarbecoviruses.

Thank you. We have revised the statement as suggested: "*LY-CoV1404 demonstrated notable neutralization potency against various SARS-CoV-2 variants, with the exception of the Omicron subvariants...*"

5) Paragraph for lines 110-122. While most readers should be familiar with the concept of SARS-CoV-2 variants and SARS-CoV (SARS1) from the scientific and lay media, most are not going to be familiar with other sarbecoviruses. Therefore, the manuscript would be improved by introducing to the names of the viruses that are being tested, the rationale for testing them and that they are SARS-CoV (SARS1) related viruses in the sarbecovirus family. Prior to showing SARS-CoV (i.e. SARS1), RS3367 and WIV1 data. Formally, the non-SARS-CoV-2 viruses are Clade 1a sarbecoviruses and SARS-CoV-2 variants are clade 1b sarbecovirus.

We express our gratitude to the reviewer for the valuable suggestion. We revised lines 110-122 as follows: "*To assess the potency and breadth of 7F3, we conducted neutralization tests against a panel of 15 pseudotyped viruses expressing the spike region of sarbecoviruses. Sarbecoviruses, encompassing a group within the Betacoronavirus genus, share genetic affinities with the Severe Acute Respiratory Syndrome coronavirus (SARS-CoV) and exhibit commonalities with other coronaviruses. Specifically, non-SARS-CoV-2 viruses, including SARS-CoV (SARS1), RS3367, and WIV1, are categorized as clade 1a sarbecoviruses, whereas SARS-CoV-2 variants fall under clade 1b sarbecoviruses.*"

6) Lines 117-120. Please provide information in the manuscript on what the specific test for synergy was performed for the data referred to in figure 1d. Was it a formal synergy calculation? If so, please provide the details in the text and method section. If

it was simply the combination experiment without a formal synergy calculation they cannot claim a test of synergy. That said, one can occasionally make a claim of potential synergy when one notes a large (i.e., >10-fold) improvement in neutralization over that of the most potent parental antibody in a combination. This sentence and elsewhere about synergy in the discussion should be revised appropriately.

We appreciate the reviewer for the correction. Lines 117-120 involve a combination experiment without a formal synergy calculation. Accordingly, we have revised the manuscript to accurately reflect this aspect, as following: “*An exploration into the combined breadth and potency of 7F3 and GW01 in a 1:1 ratio for in neutralizing SARS-CoV-2 strains and Omicron subvariants but could not observe any increase in neutralization (Fig. 1d)*”.

7) Section starting at lines 123. Binding modes for RBD targeting antibodies are generally discussed in the literature in terms of the Barnes classification schema. This schema classifies antibodies by the position of RBD required for binding (i.e., Up or both Up and Down; also known as open or open/closed, respectively) and whether or not their epitope includes the ACE2 binding site (i.e. RBM). It should be noted that Class III antibodies do not bind in the RBM but can sterically block ACE2 binding. The manuscript would be greatly improved by discussing the binding of 7F3 and GW01 in terms of this classification schema and then comparing the binding modes to other similarly classified antibodies. This will enable the reviewers and readers to better interpret the data and their significance. For example it appears from the graphics that 7F3 may be a Class I and GW01 may be class III but this reviewer is not clear without more information from the authors. If that is the case that the better comparison antibodies for 7F3 would be another class I mAbs that has a similar epitope and for GW01, LY-CoV1404 and S309 would be good comparators.

We thank the reviewer for the suggestion. We added a paragraph discussing the binding of 7F3 and GW01 in terms of this classification in the discussion section.

“Antibodies directed towards the RBD can be classified into four general categories (classes I to IV) based on their competition with the ACE2 and their recognition of the up or down state of the three RBDs in S²⁸. S309, which binds outside of the ACE2-binding site, belongs to class III antibodies. The binding sites of 7F3 coincide with the ACE2 binding site and it recognizes both up and down states of the three RBDs²⁹. GW01 binds outside RBM but it blocks ACE2 binding through steric hindrance and only binds to up RBD, as elucidated in the previous report²⁴. Thus, 7F3 and GW01 are categorized as class II and I antibodies, respectively. “

8) Line 129-131. This sentence interprets the data in Figure 1f as meaning that 7F3 binds within the RBM (i.e., ACE2 binding site). However, an alternate hypothesis that one can draw from the data in 1f is that 7F3 might sterically block ACE2 from binding to the RBM. Structural analysis is necessary to distinguish between these two

possibilities. This is done later in the paper and can be concluded only after the structural data is shown/discussed. Please modify the manuscript appropriately.

We thank the reviewer for the suggestion. Acknowledging the potential for 7F3 to sterically impede ACE2 binding to the RBM, we have incorporated this possibility into lines 129-131 accordingly in the manuscript as *“These data indicate that 7F3 might bind within the RBM region or sterically block ACE2 from binding to the RBM.”* .

9) Line 138-141. This states that, “These findings suggest that 7F3 epitope likely encompasses multiple amino acid residues within the RBD, given its retention of binding to all single and triple mutants tested. 7F3 targets a conserved, functionally critical epitope within the RBM that distinguishes it from other characterized antibodies.”

a. In lines 138-140, it is not clear what the significance of encompassing multiple amino acids it because by definition all antibody epitopes encompass multiple amino acid residues. Please revise and clarify what is intended.

b. It is not necessarily true that the epitope needs to be conserved or functionally critical. For example, the paratope of the antibody may simply accommodate side the new side chains from mutated residues while maintaining contacts to the RBD. Similarly, just because a residue has not mutated yet, does not mean that it is critical. It may simply be that we have not yet observed it yet. This is illustrated by the continued appearance of mutations at different locations in the Spike of SARS-CoV-2 and the dozens of new mutation found in the recently identified BA.2.86. Mutations may also impact the relative open or closed (i.e., up/down) state of spike or create a conformational change in RBD that modify the structural binding area of 7F3. These changes might occur outside of the binding site of 7F3 and therefore be indirect.

c. To support the conclusion the mutations are in a functionally critical region of the spike RBD or even the RBM, they would not only need to show data that the mutations significantly decrease ACE2 affinity for spike but also that viral titers are significantly decreased relative to viruses containing the parental spike proteins.

We acknowledge the reviewer's observation that the interpretation in lines 138-141 was not suitable. The retention of binding of 7F3 to all single and triple mutants tested may be attributed to these mutants not representing the epitope of 7F3. Upon comparing the epitope identified by cryo-EM and site-directed mutagenesis, it became evident that the tested epitope differed. Accordingly, we have revised the interpretation as follows: *“These findings suggest that 7F3 targets a unique epitope within RBD that distinguishes it from other characterized antibodies, which may enable its broad and potent activity against circulating and emerging SARS-CoV-2 variants.”*

10) Line 162 states that 7G-Fc is “less efficient”. This is subjective. Is it less potent neut, binding both, something else (i.e., kinetics, neut breadth)? Please be specific so a judgement can be made on the validity of the statement.

We appreciate the reviewer's feedback. We revised line 162 to be more specific in the text with the following statement “*The bispecific antibody 7G-Fc, which has a reverse orientation of GW01 and 7F3, was found to be less efficient in binding Omicron RBD proteins than G7-Fc, with lower binding affinity and bigger equilibrium dissociation constant (KD) values (Fig. 2d and 2e)*”.

11) Section on beginning on Line 182 on the prophylactic and therapeutic efficacy of G7-Fc. The BALB/C model is not the model of choice for testing antibody therapies for SARS-CoV-2. The preferred model is the Syrian golden hamster model. In addition, even amongst mouse models, it is not the best choice as it uses the mouse ACE2 receptor. When a mouse model is used, the best is the human ACE2 mouse knock-in model. Finally, there are only 4 animals in each group (standard is 10) and there is only a 1 log difference in peak viral lung titer and the level of detection limit. This does not provide a robust measure of this antibody's potential efficacy. Taken together this severely impacts the significance of the animal experimental findings.

In response to the valuable recommendation provided by the reviewer, we executed additional in-vivo protective experiments to evaluate the efficacy of G7-Fc against SARS-CoV-2 XBB.1 infection in K18-hACE2 mice. Each experimental group comprised five mice, and the mean viral lung titer of the control group exhibited a 4.6 log difference compared to the detection limit. The outcomes, as illustrated in Figure 3b, unequivocally showcase the remarkable efficacy of the antibody in conferring protection against SARS-CoV-2 XBB.1 infection in K18-hACE2 mice.

12) Section beginning on Line 197 on the Cryo-EM structure

a. The binding should be put in the context of the Barnes classification and compared use to the functional and biophysical experiments previously shown in the paper. Does the structure support and/or explain the results? That is please explain how the structure supports the proposed mechanisms of action. How does binding compare to other similarly classified RBD targeting mAbs. This is done to some extent for LY-CoV1404 but this is a better comparator for GW01 not 7F3. Also, does S309 show more similarity to GW01 than LY-CoV1404?

Thanks for pointing this out. The binding sites of 7F3 coincide with the ACE2 binding site and it recognizes both the up and down states of the three RBDs. GW01 binds outside RBM but it blocks ACE2 binding by steric hindrance and only binds to up RBD. Thus, according to the Barnes classification, 7F3 and GW01 are categorized as class II and I antibodies, respectively. S309 binds outside of the ACE2-binding site and belongs to class III antibodies. We includes an image to show the epitopes of GW01, S309 (PDB:7TLY) and LY-CoV1404 (PDB: 7MMO) on RBD (see below).

Fig: Comparison of GW01, S309 and LY-CoV1404 epitopes on RBD.

b. Based upon the neutralization data and affinity data in Figure 1, one would expect that the GW01 variable fragment (Fv) would not bind to the XBB spike protein. However, the manuscript shows that the GW01 Fv domain is binding. This is surprising. There is not a convincing argument as to why this is observed. If it was simply a loss of binding surface area or a mutation that impacted binding affinity that leads to a loss or decrease in neutralization one should still expect to see binding in a kinetics assay in figure 1b/c, but they do not detect binding to XBB in those panels. This is not adequately explained.

Thanks for pointing this out. GW01 alone can bind to WT spike protein (Cell Discovery, 2022a). Apo spike trimer usually include one up-RBD. GW01 binding induced a second RBD up, resulting in two GW01 binding with WT-S trimer with 2 up-RBDs. However, GW01 fails to bind with the XBB spike because the mutations on its epitope reduced its affinity with RBD. When linked to 7F3, 7F3 binding would induce more RBDs up, allowing GW01's binding to XBB S. In our previous study, we also observed that the design of bi-specific antibody can increase the binding affinity between antibody and Spike (Cell Discovery, 2022b).

Cell Discovery, 2022a

Wang, Y. *et al.* Novel sarbecovirus bispecific neutralizing antibodies with exceptional breadth and potency against currently circulating SARS-CoV-2 variants and sarbecoviruses. *Cell discovery*. 2022;8(1):36.

Cell Discovery, 2022b

Wang, Y. *et al.* Combating the SARS-CoV-2 Omicron (BA.1) and BA.2 with potent bispecific antibodies engineered from non-Omicron neutralizing antibodies. *Cell discovery*. 2022;8(1):104.

c. Line 207-210. Please clarify in the sentence whether the binding to RBD by 7F3 and GW01 domains are from the same G7-Fc molecule, and if it is from the same molecule is it from the same or different arms. Similarly, I believe what line 209-210 is stating is

that one arm of an antibody binds one spike and the other arm of the same antibody binds the other but the wording states it is the Fc domain that does the crosslinking. This is awkward and not formally correct since the Fc domain does not bind the spike. Please provide several cartoon viewpoints in the main figure or a supplemental figure showing the binding mode observed in the CryoEM and how the binding is occurring. This is important to interpreting what is happening.

Thank you. We added a schematic diagram in Supplementary Fig. S7b and 7c to show the binding modes of G7-Fc (also see below) and the following discussion in the revised manuscript:

“Due to the distance limitation of the GS linkers, 7F3 and GW01 from one arm of the G7-Fc bind to different RBDs, with 7F3 binding to RBM and GW01 binding to a non-RBM region.”

Supplementary Fig. S7. The XBB S/G7-Fc complex formation. **(b)** Schematic diagram of G7-Fc. **(c)** Schematic diagram showing how Fc regions crosslink S trimers. The red dashed line represents a G7-Fc antibody. The zoomed-in view of RBD/G7-Fc is shown on the right panel. Due to the distance limitation of (GGGGS)₄ linker (~70Å) between GW01 and 7F3, 7F3 and GW01' bound to the same RBD were derived from two G7-Fc antibodies. Yellow spheres represent the N-terminal of 7F3. Magenta spheres represent the C-terminal of GW01/GW01'.

d. Are there structures of parental antibodies alone and/or as a mixture? If so how does

their binding compare to what is seen in the CryoEM? If there is not, at minimum, please provide negative stain EM reconstructions or low-resolution Cryo-EM to determine if the bispecific induces binding modes that are different than parental antibodies do by themselves. This is especially important in light of the finding that the GW01 Fv domain is binding spike. For example, is the binding of 7F3 inducing a conformational change in a mixture or bispecific format that then allows GW01 to bind?

Thank you. GW01 could not bind XBB-S. However, the structure of WT S in complex with GW01 was previously determined with two GW01 binding to S trimers with two up-RBD.

To answer this question, we performed negative staining EM analysis. The negative stain EM images of 7F3 IgG, GW01 IgG and G7-Fab incubated with XBB S trimers were added in Supplementary Fig. S4c (See below). XBB S trimers did not form trimer dimers when incubating with parental antibodies or G7-Fab. These results support the statement that G7-Fc synergistically induces the formation of trimer dimers.

Supplementary Fig. S4. Purification of SARS-CoV-2 XBB S in complex with G7-Fc. (a) Gel-filtration curve and SDS-PAGE of SARS-CoV-2 XBB S complexed with G7-Fc complex. For comparison, the gel-filtration curves of SARS-CoV-2 XBB S and the molecular weight standards were also shown. (b) SDS-PAGE of G7-Fc and G7-Fab. G7-Fab was obtained by digesting G7-Fc using papain. (c) Negative staining EM images of SARS-CoV-2 XBB S alone, XBB S/G7-Fc complex, XBB S/G7-Fab, XBB S/7F3 IgG, XBB S/GW01 IgG, and XBB S/7F3 IgG/GW01 IgG.

S/7F3 IgG, XBB S/GW01 IgG and XBB S/7F3 IgG/GW01 IgG, showing that only G7-Fc binding induces the formation of trimer dimer.

e. Line 207. Since each of the three RBDs on spike naturally and independently switch between up and down states it is not formally possible to know from the provided data whether binding of G7-Fc induces a 3-RBD up state or if it simply when RBD goes into an up position G7-Fc lock it in the up state while it remains bound. In the Cryo-EM experiment G7-Fc is in excess so it is more likely to captures in a steady state saturated binding with RBD bound than unbound. This data indicates that G7-Fc can bind up to 3 RBDs in the up state.

Thanks for pointing this out. In general, most apo-state S trimer includes only one up RBD. Structural superposition showed that GW01 would clash with another S protomer when 7F3 bound to the S trimer with one up RBD (Fig S7a). It is necessary that the adjacent down RBD is induced to the up state to provide sufficient space for GW01 binding. Thus, we could infer that the binding of G7-Fc induces a 3-RBD up state of S protein.

Supplementary Fig. S7. The XBB S/G7-Fc complex formation. (a) Structure superposition of XBB S-RBD/G7-Fc with apo-state S. Structures are aligned over the up RBD. GW01 clashes with the adjacent down RBD when 7F3 binds with the up RBD. (b) Schematic diagram of G7-Fc. (c) Schematic diagram showing how Fc regions crosslink S trimers. The red dashed line represents a G7-Fc antibody. The zoomed-in view of RBD/G7-Fc is shown on the right panel. Due to the distance limitation of

(GGGS)₄ linker (~70Å) between GW01 and 7F3, 7F3 and GW01' bound to the same RBD were derived from two neighboring G7-Fc antibodies. Yellow spheres represent the N-terminal of 7F3 and magenta spheres represent the C-terminal of GW01/GW01'.

f. It is very interesting that there are a large number of hydrogen bond networks being created between RBD and 7F3. This is more than is typically seen and should be explored and compared to other antibodies that bind the same region of RBD. What about salt bridges? How were the surface area and hydrogen bond contacts determined? Was PISA used? If so, please provide the PISA data as a supplemental table/figure and add it to the methods.

Thanks for this suggestion. We used PISA determine the hydrogen bond networks between 7F3 and XBB S-RBD using a 4.0 Å cutoff (consistent with the previous paper on structure of WT S-GW01 (*Wang Y et al. Cell discovery 2022a*)). According to PISA, the binding of 7F3 to XBB RBD buries 976.4 Å² surface area. 7F3 and XBB RBD form 18 pairs of hydrogen bonds and do not form salt bridges. We added these results in Supplementary Table. S3 and S4 in the revised manuscript.

g. Lines 228-231 describes a reduction in binding surface area for GW01 of 56.6 angstroms in XBB spike relative to BA.1 Spike. There is no reference provided. Also, what was the overall buried surface area in both cases? Given that there was likely >500 angstroms of buried surface area remaining on XBB, it seems unlikely by itself to explain the total loss of binding and neut for XBB in Figure 1b/c/d. Were other binding modes explored (i.e, salt bridges, charge interactions)?

Thanks. The binding of GW01 to BA.1 S-RBD buries 718.3 Å² surface area (*Wang Y et al. Cell Discovery, 2022b*). We added the results of PISA in Supplementary Table. S3 in the revised manuscript.

We compared the structure of WT S/GW01 (*Cell Discovery, 2022a*) with XBB S/GW01 and analysed the residues involved in GW01 binding. The N501Y and Y505H mutations may result in the escape of XBB. Besides, GW01 forms ten pairs of hydrogen bonds and a salt bridge with WT S-RBD. In XBB S/GW01, GW01 forms 6 pairs of hydrogen bonds and no salt bridge with XBB S-RBD (Supplementary Table. S5). The loss of these interactions may result in the deficiency of GW01 binding.

h. Line 243-245 describes the frequency of the Y501A mutation. There is no reference for the database used and searched in the methods and text. The text should also state which database was used. In addition, while the frequency of 501 alanine may be low there are significant numbers of changes at position 501 other than alanine. They should also note the other amino acids changes that are in the database at each position in the database they search. The burden is on the authors to show that the other mutations do not impact activity. In the absence of data showing the other mutations do not impact

binding and neut the assumption should be that ANY change at the residues noted in Figure 5d will disrupt G7-Fc activity.

We added the reference for the used database in the text in line 286. We added the Supplementary Fig. S8b showing the the mutation rates of other residues in 501, which are extremely low. This suggests that G7-Fc can potentially neutralize the vast majority of currently circulating SARS-CoV-2 strains.

i. Line 256-258 describes a G7-Fc epitope. Since this is a bispecific antibody it is best to refer to the individual epitopes of each Fv domain independently rather than including them as one large epitope. This is because mutations in each component Fv epitope should have independent effects on the molecule. Therefore, when discussing mutations and their effects it is best to put them in the context of the Fv domain that is being impacted based on the structure.

Thanks. We changed this statement in the revised manuscript, as following *“These results indicate that G7-Fc bispecific antibody binds to two highly conserved epitopes across variants of the Omicron lineage of SARS-CoV-2, thereby explaining the ability of G7-Fc to neutralize even the most antigenically evasive Omicron variants.”*

13) Line267-270. What data is being used to support the statement that G7-Fc synergistically induces the formation of trimer dimers? It is unclear to this reviewer what data in the manuscript supports this conclusion. I do not believe the data in figure 4 supports this conclusion. Also, there is no evidence of synergistic neutralization. Please modify the manuscript to provide a clarification of how the data supports the conclusion or remove the statement.

Thanks for pointing out this. The pseudovirus neutralization assay in Fig. 1c and 2e indicated that the G7-Fc bispecific antibody increases the neutralization potency of parental antibodies to the tested pseudoviruses compared to the individual antibodies. In addition, we performed the negative staining EM analysis (Supplementary Fig. S4c), indicating that XBB S trimers did not form trimer dimers when incubating with parental antibodies 7F3 IgG and GW01 IgG, or G7-Fab. According to the cryo-EM structure of XBB S/G7-Fc, 7F3 and GW01 bind to two different sites in RBD, enlarging the binding area and improving the neutralization potency of the bispecific antibody. These biochemical experiments and structure data suggest the synergistic neutralization mechanism of G7-Fc. We changed the following discussion in the revised manuscript:

“Therefore, the bispecific antibody G7-Fc inhibits SARS-CoV-2 XBB variant infection by synergistically inducing the formation of trimer dimers. Both 7F3 and GW01 bind to highly conserved epitopes in XBB, enlarging the interface area, thereby improving the affinity between the RBD and single scFv and blocking the RBD from interacting with the ACE2 receptor.”

14) Line 295-299. What data supports the finding that the binding of G7-Fc induces conformational change that allows GW01 to bind. I did not see experiments showing this in the manuscript. High-resolution structures of the unbound spike, spike with each parental mAb alone and spike with mixtures of the two mAbs. Both XBB and a spike that is known to bind both GW01 and 7F3 are needed to be compared and identify a conformational change. In addition, an order of addition experiment using BLI that shows that the binding of 7F3 first allows GW01 to bind is needed. Without these data, the claim should be removed.

As shown in Fig. 1d, the mixture of 7F3 and GW01 did not increase their neutralization activity compared to the single antibody. Therefore, addition of 7F3 to the RBD might not increase the binding of GW01 to RBD. 7F3 only increases GW01 binding to the RBD in the scFv format within the bispecific antibody construction.

15) Figure 1d: Please indicate what the ratio of 7F3 and GW01 were in the combination neutralization assay.

We appreciate the reviewer's suggestion. The ratio of 7F3 and GW01 were 1:1 in the combination neutralization assay. We added this in the figure legend of figure 1c.

16) Figure 1e/f. It is not clear what the order of addition is for the figures. Also, please provide a percent competition value as a table in this figure or in the supplement that clearly indicates what the competitor is and what the analyte is. One would expect ACE2 self-competition to be near 100% it appears to be <50%, which is very weak and would indicate that this experiment is not interpretable. Please provide the table and if self-competition is < 75% it needs to be repeated.

Sorry for the confusion. We repeated the assay using ACE2+ VRC01 IgG isotype as a positive control instead of ACE2 only. Percent competition values were calculated based on the binding area under the curve. Binding curve of ACE2+ VRC01 IgG isotype was employed as a positive control and appeared to be 100% competition value, while the binding curve of mAb VRC01 and RBD served as a negative control and was designated as a 0% value. We added the percentage of competition values in figure 1d.

17) Figure 2f: Because there are more virus variants tested in this table than in figure 1d, this data table should include a comparison to the parental Abs and the mix of the parental antibodies as controls.

We included the neutralization results of parental Abs and the mix of the parental antibodies in figure 2e for comparison.

Other comments:

1) Line 81-82: This sentence says the rapid spread of SARS-CoV-2 variants make it

urgent to develop pan-sarbecovirus antibodies . However, it is not clear to this reviewer how the appearance of SARS-CoV-2 variants means we need to have better SARS-CoV (SARS1) like antibodies. I recommend adding the logic or modifying the sentence.

We thanks the reviewer for the comment. We modified the paragraph as follows “*With the emergence and rapid dissemination of SARS-CoV-2 variants, the imperative for broadly neutralizing antibodies capable of pan-sarbecovirus neutralization has intensified. Antibodies possessing extensive neutralization across various sarbecoviruses are more likely to target conserved epitopes, rendering them more resilient against immune evasion by swiftly emerging SARS-CoV-2 variants. Consequently, the development of novel pan-sarbecovirus antibodies exhibiting broad and potent activity has become an urgent necessity for both the prevention and treatment of coronavirus disease 2019 (COVID-19). Moreover, considering that coronaviruses have incited three pandemics within the past two decades, involving SARS-CoV, MERS-CoV, and SARS-CoV-2, the demand for pan-sarbecovirus antibodies becomes paramount in anticipation of potential recurrences of coronavirus pandemics.*”

2) Line 104. The generic name bebtelovimab should be mentioned with the first use of LY-CoV1404 earlier in the paper on line 66. Similarly, earlier Evusheld should have the generic names provided.

We modified the sentence as suggested.

3) Line 106. The manuscript would be improved by providing a few more details about GW01 such as where it binds, is it a human antibody. It says it is a sarbecovirus Nab. So a reader may want to know if it was isolated to target SARS-CoV (SARS1) and then found to have SARS-CoV-2 activity or was it isolated against SARS-CoV-2 and found to have SARS1 breadth?

GW01 was isolated from a COVID-19 recover patient against SARS-CoV-2 and found to have SARS-CoV breadth, targeted the epitope outside of RBM and inhibit RBD binding to ACE2. We added the details in line 117 as following: “*We previously isolated a broadly NAb from a COVID-19 convalescent, named GW01²⁴, which targeted the outside of RBM and displayed broadly neutralization potency against SARS-CoV-2 and SARS-CoV. However, it failed to bind BQ.1.1, XBB.*”

4) Line 107-109, indicates the affinities in Fig 1b and 1c accord with ELISA results from 1a. Since multiple approaches can be used to determine binding (and affinity), a reader may appreciate mentioning in lines 101-103 that the binding there is ELISA and then say that the affinities were measured using biolayer interferometry in lines 107-109.

Thanks. We modified the sentence as suggested.

5) Section on construction of bispecific antibodies starting at line 144. Since there are many formats and types of bispecific antibodies, the manuscript would be improved by a short description of the bispecific antibody format being used. Of particular note, this is a tetravalent bispecific. Does it contain Fc mutations (i.e., knob and hole mutations, Fc-effector function mutations, etc)? What is the order of the variable fragments on each arm in each version of the molecule. For this last item, a graphic showing both versions being made and tested should be provided in the main figure or a supplemental figure.

Thank you for the suggestions. IgG-like bispecific antibodies were constructed by connecting the scFvs of two non-Omicron neutralizing antibodies with a (G4S)4 linker and fusing to the Fc region as shown in Fig.2a (See below). We added the details in the manuscript in line 175, as following: “*Briefly, the single-chain variable fragments (scFv) of antibodies was linked with a (Gly4Ser)4 linker, and subsequently fused to IgG1 Fc. The sequential arrangement of antibody G7-Fc, from N to C terminus, is delineated as follows: GW01 VL-(Gly4Ser)3-GW01 VH-(Gly4Ser)4-7F3 VL-(Gly4Ser)3-7F3 VH-Hinge-CH2-CH3.*”

Fig.2a. Schematic diagrams of the structures of bispecific antibodies.

6) Line 164 states the order increases the binding affinity. This is not formally correct. The order impacts the binding affinity. Please change the wording to either say having X first increases the affinity or that the order impacts the affinity. Also, they may want to discuss the binding relative to the neutralizing activity.

We modified the word “increase” to “impact” as suggested.

7) Line 170-172. Why is it surprising that G7-Fc neutralizes the viruses panel shown? Based on the parental antibody neutralization and combination experiment, one would expect it to neutralize as it does. So it would be surprising if it did not. Please rephrase.

We added the parental antibodies GW01 and 7F3 in parallel with G7-Fc in figure 2e. The parental antibodies GW01, 7F3, and their combination only neutralized 32.1%, 67.9%, and 89.3% with GM IC50 values of 1.04, 2.51 and 4.26 nM. While G7-Fc 100% neutralized the 28 pseudovirus panel with GM IC50 as low as 0.13 nM. We added these descriptions in the text.

8) Line 343. Since the molecules being tested are not Fabs but mAbs, which are bivalent, or bispecific, which are tetravalent, a 1:1 binding model are typically not the correct binding model to use for kinetic (affinity) analyses. For a standard mAb with an Fc domain a bivalent binding model is most appropriate to account for inherent avidity. For a tetravalent molecule it is not clear what model is best to use but a bivalent model appears more appropriate than a 1:1 binding model.

We thank the reviewer for the great suggestion. We have changed the data analysis using 1:2 bivalent binding model for both standard mAbs and bispecific antibody affinity analyses, the reanalyzed data is updated in the figure 1b and figure 2d.

9) Competition assay by BLI method. This method does not explain the ACE2 binding competition assays. This is critical because it is not clear what order the assay is being performed in figure 1 and 4.

We thank the reviewer for the suggestion. We added the method details of ACE2 binding competition assay in the method section as following:

ACE2 competition assay by BLI

As for ACE2 competition binding assay, 600 nM antibody was incubated with 100 nM RBD-8his protein for 30 mins earlier. Procedure is similar to Ab competition assay. First, ACE2-Fc was captured by AHC biosensor for 600s. Second, after baseline 120s in PBST buffer, the sensor were blocked with IgG1 isotype control for 600s. Third, the sensor was soaked into the antibody-RBD pre-mixture for 600s after baseline again in buffer; Mixture of ACE2-Fc and RBD was as a positive control, and mixture of isotype mAb VRC01 and RBD was as a negative control.”

10) The graphs in figure 1b and 2d are too small to see. I suggest that all or only a select subset be included in the figure and the remaining be placed into a supplement to allow better visualization of the graphs.

Thanks for the suggestion. We moved figure 1b and 2d to supplement Fig.S2 and Fig.S3,

respectively, to allow better visualization of the graphs.

11) Figure 1d and 1f. The dark fonts in red/green colored boxes are difficult to read. Suggest changing to a light-colored font in dark boxes. In addition, figure 1d is not color blind friendly with green next to red boxes.

We have improved the figure qualities of figure 1c and figure 2e and made them more friendly to the readers.

12) The legend description for panel (f) is shown as (e)

We revised this as suggested.

13) Panel 2g, y-axis is labeled Inhibition rate. It is not a rate. It should be labeled as percent inhibition or neutralization percent depending on how the calculation was performed.

Thanks for your points. We have revised the figure as suggested in figure 2f.

14) Figure 3b. To provide better identification of the measured values relative to one another, please provide minor tick marks for the y-axis and provided the log value for the lower limit of detection in the legend.

Thanks for your suggestion. We have revised the figure as suggested in figure 3b.

15) It is difficult to figure out what is 7F3, GW01 and RBD in the models of the upper left-hand panel of figure 1a . Please label each.

We labeled the RBD in Fig.1a.

16) It is difficult to see the yellow outline of the ACE2 binding site in panel 1C, please make more prominent and consider changing the color to make it more visible.

Thanks for your suggestion. We used the red outline of the ACE2 binding site in Fig. 4c to make it more visible.

Fig 4c. Surface representation of RBD showing the buried binding site, including GW01(orange), 7F3 (dodger blue), and the red line indicates the ACE2 binding site.

17) The manuscript switches between up and open to describe RBD's state. My preference is to cue the reader that open equals up and closed equals down. And then to stick with one or the other nomenclature for the remaining part of the article.

Thanks for pointing this out. As suggested, we changed "open" to "up" in the revised manuscript:

"line 45: Two scFvs of G7-Fc synergistically induced XBB S trimers to the state with all three RBDs up and inhibited RBD binding to ACE2."

"line 248: The binding of G7-Fc induces S trimers to the state with all three RBDs up."

"line 352: The structure of XBB S/G7-Fc complex indicates that the binding of 7F3 induces all three RBDs to up state."

REVIEWER COMMENTS

Reviewer #1 (Remarks to the Author):

Wang et al. present a bispecific antibody, G7-Fc that is able to neutralize many SARS-CoV-2 variants by interacting with conserved residues on the spike. Their cryo-EM structure of the complex shows that the bispecific antibody can crosslink the spike by forming a head-to-head dimer of trimers. The bispecific antibody has also shown promise in mice. Such a bispecific would be useful in the face of the continually mutating SARS-CoV-2.

All comments have been satisfactorily addressed and their results merit publication.

Reviewer #2 (Remarks to the Author):

The authors have done an excellent job responding to the critiques and suggestions. I have no further comments. The manuscript is much clearer in this form.

Reviewer #3 (Remarks to the Author):

Many of the concerns of the reviewer have been addressed. However, there remains some concerns need to be addressed.

Below are numbered by the original concern #

2. Thank you for providing the method and gating strategy. However, this method and notable portions of the methods section do not have the sources of the reagents used and should be added.

3. Thank you for converting much of the data to Molar. Panels 1a and 2c still need to be converted.

7. This concern is not fully addressed. The proposed paragraph that has been added is not sufficient for several reasons. First, its placement in the discussion does not provide the reviewers and readers context of how interpret results compared to other similarly classified antibodies until the discussion. Placement of the paragraph would be best if it was placed near at the point where binding modes and receptor binding motif (RBM) are first being discussed for the mAbs in this paper. Second, data that has been highlighted and added in the revision, shows that GW01 only binds to RBD when it is in the up position and that its epitope lies outside of the RBM. This means that GW01 is a class IV antibody. The paragraph incorrectly interprets that ACE2 blockade by GW01 makes it a class I antibody. However, it is binding to the RBM that matters not ACE2 blockade for assigning class. For example, LY-CoV1404 is a class III antibody that blocks ACE2 binding. There are other similar examples in the literature. This should be corrected. Lastly, the original request to use the classification scheme to enable comparison to other similarly classified antibodies is generally not done. For example, there is a comparison to S309 and LY-CoV1404 but they are class III. However, neither mAb in the paper is a class III mAb. Despite GW01 being a class IV mAb, there are no other class IV mAbs in the paper. These exist in the literature and therefore should be used as comparators in figures and the discussion. Similarly, mAb 7F3 is the same class as LY-CoV555, class II. There are other class II mAbs that are broad and potent. What are the differences between binding of 7F3 and LY-CoV555 (and others) that allow it to overcome mutations in spike that abrogate or severely limit the binding of other class II mAbs. These are important comparisons that will help elucidate the mechanisms that allow antibodies to overcome resistance of current variants to antibody therapeutics.

9. There are now antibodies that are published or in preprints that bind and neutralize. Therefore, it is better to have the interpretation focus on the uniqueness from antibodies in this manuscript rather than all published antibodies. I suggest modifying the sentence further to say, "These findings suggest that 7F3 targets an epitope within RBD that distinguishes it from the other characterized antibodies we investigated, and enables its broad and potent activity against circulating and emerging SARS-CoV-2 variants."

10a. Now that Barnes classification is understood, GW01 comparison to another class IV and 7F3 comparison to another class II would be best. Since LY-CoV555 is not as broad as other class II mAbs, I suggest using a broader class II mAb.

10b. If the mechanism that was proposed in the response is correct, then mixing Fab of 7F3 with XBB spike at saturating concentrations and followed by addition of Fab of GW01 prior to making an EM grid, should results in at least some of the class averages having evidence of GW01 bound. Alternatively, a BLI binding experiment may have similar results. Specifically, if Fab of 7F3 is first allowed to saturate binding of XBB spike prior to dipping the sensor tip into a solution containing

both Fab of 7F3 and Fab of GW01, one should see a further increase in the binding signal. Without experiments like these, the observations may simply be due the Fv of GW01 being in close proximity to its expected epitope (i.e., the binding observed for GW01 is not high affinity due to RBD being in the up position but rather is because the Fv is effectively at a higher local concentration).

10e. This clarification of the binding modes and epitopes presented in the modified figure and response are helpful. The point that there is not sufficient space for binding of GW01 to bind when the adjacent RBD is in the down position is a critical point that needs to be made more prominently.

An important implication of this relates to understanding the synergy mechanism. That is, in order for the first G7-Fc to bind, 2 RBDs must be in the up position. To have the 2nd RBD bound, the third RBD must be in the up position. Once the second RBD is bound the third G7-Fc can immediately engage without waiting. In this model, each binding event “primes” the binding for the next event and could be a major component of the synergy that is being observed.

In the classes that were observed, were there any that showed 2RBD up with only 1 or 2 of the RBDs bound by G7-Fc?

10i. The modified manuscript claims that the mAb should work for all variants based on GSAID data and 95% conservation on a limited set of positions. Is that true across all of the PISA positions? Please show a table with the conservation at each position or a heat map on the RBD for the PISA positions. This will visually show this better and clarify the probability of conservation across all contacts instead of a limited number.

I also appreciate the modified sentence. However, even though they are independent, it is probably more accurate to say “two complementary epitopes” rather than “two highly conserved epitopes”.

14. Thank you for the response.

Models of synergy for antibodies include Zhou (Science. 2022. PMID:35324257) which proposed a 2-step models for antibody synergy where binding of the first antibody locks RBD into the up position and enhances the binding of the second antibody. For multispecifics, the Zhou model and crosslinking have been observed (Sci Transl Med. 2021. PMID:34519517 and bioRxiv. 2022. PMID:35982683)). This paper proposes synergy is due to increase surface area from GW01 binding. But given the crosslinking of spikes observed in Cryo-EM, this is likely a major mechanism and should be investigated. To determine the relative role of crosslinking to the increased potency observed vs

the role of GW01 binding in the bispecific, binding and neutralization of a G7-Fc configured molecule where the GW01 component was replaced with a genetically comparable scFv that targets a different virus should be made and tested. This would discriminate between the two possible hypotheses and should be performed. Once the mechanism is determined, it should be compared to that of published models for synergy for monoclonals and multispecifics.

16. Thank you for adding the method. It is unusual to use area under the curve to calculate percent competition. Most groups use maximum binding of the analyte relative to its expected binding. Please add the similar calculation method the antibody competition experiments.

Other comments section numbering

1. Thank you for modifying the text. MERS-CoV has not caused a pandemic.

REVIEWER COMMENTS

Reviewer #1 (Remarks to the Author):

Wang et al. present a bispecific antibody, G7-Fc that is able to neutralize many SARS-CoV-2 variants by interacting with conserved residues on the spike. Their cryo-EM structure of the complex shows that the bispecific antibody can crosslink the spike by forming a head-to-head dimer of trimers. The bispecific antibody has also shown promise in mice. Such a bispecific would be useful in the face of the continually mutating SARS-CoV-2.

All comments have been satisfactorily addressed and their results merit publication.

We are grateful for your summary of the manuscript and your encouraging comments.

Reviewer #2 (Remarks to the Author):

The authors have done an excellent job responding to the critiques and suggestions. I have no further comments. The manuscript is much clearer in this form.

We appreciate the positive feedback from the reviewer.

Reviewer #3 (Remarks to the Author):

Many of the concerns of the reviewer have been addressed. However, there remains some concerns need to be addressed.

Below are numbered by the original concern #

2. Thank you for providing the method and gating strategy. However, this method and notable portions of the methods section do not have the sources of the reagents used and should be added.

We added the sources of the reagents used in the method section in line 417-421.

3. Thank you for converting much of the data to Molar. Panels 1a and 2c still need to be converted.

Thank you. We converted the antibody concentration to molar units in the Panels of Fig 1a and 2c in the revised manuscript.

7. This concern is not fully addressed. The proposed paragraph that has been added is not sufficient for several reasons. First, its placement in the discussion does not provide the reviewers and readers context of how interpret results compared to other similarly classified antibodies until the discussion. Placement of the paragraph would be best if it was placed near at the point where binding modes and receptor binding motif (RBM) are first being discussed for the mAbs in this paper. Second, data that has been highlighted and added in the revision, shows that GW01 only binds to RBD when it is in the up position and that its epitope lies outside of the RBM. This means that GW01 is a class IV antibody. The paragraph incorrectly interprets that ACE2 blockade by GW01 makes it a class I antibody. However, it is binding to the RBM that matters not ACE2 blockade for assigning class. For example, LY-CoV1404 is a class III antibody that blocks ACE2 binding. There are other similar examples in the literature. This should be corrected. Lastly, the original request to use the classification scheme to enable comparison to other similarly classified antibodies is generally not done. For example, there is a comparison to S309 and LY-CoV1404 but they are class III. However, neither mAb in the paper is a class III mAb. Despite GW01 being a class IV mAb, there are no other class IV mAbs in the paper. These exist in the literature and therefore should be used as comparators in figures and the discussion. Similarly, mAb 7F3 is the same class as LY-CoV555, class II. There are other class II mAbs that are broad and potent. What are the differences between binding of 7F3 and LY-CoV555 (and others) that allow it to overcome mutations in spike that abrogate or severely limit the binding of other class II mAbs. These are important comparisons that will help elucidate the mechanisms that allow antibodies to overcome resistance of current variants to antibody therapeutics.

Thank you for your valuable feedback. We apologize for the misclassification of the antibodies. We moved this paragraph from “discussion” to the “result” section and rewrote this part.

We have now included BD23¹, classified as class II, for comparison with 7F3. Additionally, CR3022², categorized as class IV, was utilized as a comparison to GW01. To discern epitope differences, we conducted an antibody competition binding assay by BLI. We added the results in Figure 4b-c (also see below) and in the 2nd paragraph of Page 10 as follows:

*“Antibodies directed towards the RBD can be classified into four general categories (classes I to IV) based on their competition with the ACE2 and their recognition of the up or down state of the three RBDs in spike. 7F3 strongly competed with the class II antibody BD23 for WT RBD engagement (100%, **Fig. 4b**). In contrast, CR3022 and GW01 showed no competition or limited competition with 7F3 (0% and 17.4%, respectively). Moreover, CR3022², class IV, exhibited strong competition to the WT RBD compared to GW01 (96.21%, **Fig. 4c**), whereas 7F3 and BD23 showed no competition or limited competition with 7F3 (0% and 34.81%, respectively, **Fig. 4c**). These findings*

suggest a substantial epitope overlap exists between 7F3 and BD23, and overlap between CR3022 and GW01 epitopes. Based on the Barnes Classification of antibodies, 7F3 is categorized as a class II antibody, whereas GW01 is classified as a class IV antibody.”

Fig. 4. (b) Binding of 7F3 to the SARS-CoV-2 RBD in competition with CR3022 (blue), GW01 (pink), and BD23 (black) was assessed using BLI (left). (c) Binding of GW01 to the SARS-CoV-2 RBD in competition with 7F3 (red), BD23 (black), and CR3022 (blue) was evaluated using BLI (right).

- 1 Cao, Y. *et al.* Potent Neutralizing Antibodies against SARS-CoV-2 Identified by High-Throughput Single-Cell Sequencing of Convalescent Patients' B Cells. *Cell* **182**, 73-84.e16, doi:<https://doi.org/10.1016/j.cell.2020.05.025> (2020).
- 2 Yuan, M. *et al.* A highly conserved cryptic epitope in the receptor binding domains of SARS-CoV-2 and SARS-CoV. *Science* **368**, 630-633, doi:10.1126/science.abb7269 (2020).

Furthermore, we added two new figures (Supplementary Fig. S8 and S9) to compare the binding modes of antibodies within one class. The results were included in the last paragraph of Page 10 and the 2nd paragraph of Page 11 as follows:

“Structure comparison reveals that the binding modes of XBB S/7F3 and WT S/BD23 (PDB ID: 7BYR) are quite similar. Other than that, 7F3 was implicated in more interactions with RBD residues, and the E484A, F486S, F490S, Q498R, N501Y, and Y505H mutations may interfere with the contacts between BD23 and SARS-CoV-2 variants (Supplementary Fig. S8).”

“Comparison of XBB S/GW01 and WT S/CR3022 (PDB ID: 6W41) reveals that the CR3022 epitope, which primarily consists of L368-F392 residues, is located near the bottom of the RBD. The S371F, S375F, T376A and R408S mutations may cause the loss of binding affinity of CR3022 to SARS-CoV-2 variants (Supplementary Fig. S9).”

Supplementary Fig. S8. Comparison between 7F3 and BD23. (a) Structure comparison between XBB S/7F3 and WT S/BD23 (PDB ID: 7BYR). Structures are aligned on RBD. (b) Sequence alignment of SARS-CoV-2 WT, Delta, BA.1, BA.5, BQ.1.1, XBB, XBB.1.5 and XBB.1.16. Residues involved in XBB S/7F3 are marked with triangles in blue. Residues involved in WT S/BD23 are marked with triangles in pink.

Supplementary Fig. S9. Comparison between GW01 and CR3022. (a) Structure comparison between XBB S/GW01 and WT S/CR3022 (PDB ID: 6W41). Structures are aligned on RBD. (b) Sequence alignment of SARS-CoV-2 WT, Delta, BA.1, BA.5, BQ.1.1, XBB, XBB.1.5, and XBB.1.16. Residues involved in both XBB S/GW01 and BA.1 S/GW01 are marked with triangles in orange. Other residues involved in BA.1 S/GW01 are marked with triangles in purple. Residues involved in WT S/CR3022 are marked with triangles in green.

9. There are now antibodies that are published or in preprints that bind and neutralize. Therefore, it is better to have the interpretation focus on the uniqueness from antibodies in this manuscript rather than all published antibodies. I suggest modifying the sentence further to say, “These findings suggest that 7F3 targets an epitope within RBD that distinguishes it from the other characterized antibodies we investigated, and enables its broad and potent activity against circulating and emerging SARS-CoV-2 variants.”

Thank you for your suggestions. We have modified the sentence as suggested in 163-164 of the revised manuscript.

10a. Now that Barnes classification is understood, GW01 comparison to another class IV and 7F3 comparison to another class II would be best. Since LY-CoV555 is not as broad as other

class II mAbs, I suggest using a broader class II mAb.

Thank you. We added class II mAb BD23 and class IV mAb CR3022 as comparison to show the epitope similarity (Fig. 4 b, c). The detailed results are shown in Fig. 4b-c.

10b. If the mechanism that was proposed in the response is correct, then mixing Fab of 7F3 with XBB spike at saturating concentrations and followed by addition of Fab of GW01 prior to making an EM grid, should results in at least some of the class averages having evidence of GW01 bound. Alternatively, a BLI binding experiment may have similar results. Specifically, if Fab of 7F3 is first allowed to saturate binding of XBB spike prior to dipping the sensor tip into a solution containing both Fab of 7F3 and Fab of GW01, one should see a further increase in the binding signal. Without experiments like these, the observations may simply be due the Fv of GW01 being in close proximity to its expected epitope (i.e., the binding observed for GW01 is not high affinity due to RBD being in the up position but rather is because the Fv is effectively at a higher local concentration).

We appreciate your feedback and have conducted the BLI binding experiment as suggested to elucidate the mechanism of the bispecific antibody. Supplementary Figure S11 illustrates the experimental setup and results of the antibody fragment (Fab) binding to the XBB trimer as measured by BLI. In brief, the XBB trimer was immobilized onto the NTA biosensor. The sensor tip was then exposed to GW01 Fab or VRC01 Fab isotype control, followed by 7F3 Fab. The results indicated no discernible increase in binding signal compared to the VRC01 Fab isotype control. Similarly, when the biosensor tip was first exposed to 7F3 Fab followed by GW01 Fab, no binding signal increase was observed. Additionally, XBB S trimers failed to form trimer dimers when incubated with G7-Fab (Supplementary Fig. S4b-c). The IgG1 format of the bispecific antibody is essential for inducing the formation of trimer dimers.

These findings suggest that G7-Fc inhibits SARS-CoV-2 XBB variant infection by synergistically inducing the formation of trimer-dimers. Both 7F3 and GW01 bind to the complementary epitopes in XBB, enlarging the interface area, thereby improving the affinity between the RBD and single scFv and blocking the RBD from interacting with the ACE2 receptor. The structural arrangement of the bispecific antibody, particularly the Fc region, plays a critical role in facilitating G7-Fc binding to the trimer, representing one of the mechanisms of antibody neutralization.

Supplementary Fig. S11 Illustration of 7F3 and GW01 Fab binding to XBB trimer as measured by BLI. (a) The sensor tip immobilized with XBB protein was first exposed to GW01 Fab or VRC01 Fab isotype control for binding, followed by interaction with 7F3 Fab, as depicted in the figure above. **(b)** The sensor tip immobilized with XBB protein was initially exposed to 7F3 Fab or VRC01 Fab isotype control for binding, followed by interaction with GW01 Fab.

3 Burton, D. R. Antiviral neutralizing antibodies: from in vitro to in vivo activity. *Nat Rev Immunol* **23**, 720-734, doi:10.1038/s41577-023-00858-w (2023).

10e. This clarification of the binding modes and epitopes presented in the modified figure and response are helpful. The point that there is not sufficient space for binding of GW01 to bind when the adjacent RBD is in the down position is a critical point that needs to be made more prominently.

An important implication of this relates to understanding the synergy mechanism. That is, in order for the first G7-Fc to bind, 2 RBDs must be in the up position. To have the 2nd RBD bound, the third RBD must be in the up position. Once the second RBD is bound the third G7-Fc can immediately engage without waiting. In this model, each binding event “primes” the binding for the next event and could be a major component of the synergy that is being observed.

In the classes that were observed, were there any that showed 2RBD up with only 1 or 2 of the RBDs bound by G7-Fc?

Thanks for pointing this out. The 2D and 3D classifications observed only the state of 3 RBDs up (see the figure below). This may be due to the sufficiently long incubation time, which caused G7-Fc to induce all three RBDs up. “*The XBB S trimer was mixed with G7-Fc in a 1:1.3 molar ratio, incubated at 4°C for 0.5 h, and further purified by gel filtration chromatography on a Superose 6 increase 10/300 column (GE Healthcare). The peak fraction was concentrated to 0.5 mg/ml in 20 mM Tris, pH 8.0, and 200 mM NaCl for cryo-EM study.*”

2D classification

3D classification

mask monomer1 3D class

mask monomer2 3D class

We agree that it is very possible to observe a state with 2 RBDs up if we incubate less time. In our previous research, we observe 6 different states showing the stepwise binding of a GW01-16L9 bi-specific antibody when the antibody was incubated with S

trimer for 10 mins before freezing for the Cryo_EM study (See below).³

Fig. 3 Cryo-EM structures of the Omicron S trimer in complex with the bispecific antibody FD01 IgG. Cryo-EM structure of the prefusion stabilized SARS-CoV-2 Omicron S ectodomain trimer in complex with the bispecific antibody GW01-16L9 (FD01) IgG, revealing 6 states of the complex. State 1: up-down-down RBDs, 1 scFv, 3.47 Å. State 2: up-up-down RBDs, 3 scFvs, 3.70 Å. State 3: up-up-half-up RBDs, 4 scFvs, 3.91 Å. State 4: up-up-up RBDs, 4 scFvs, 3.47 Å. State 5: 3-up RBDs, 6 scFvs, 3.87 Å. State 6: trimer dimer, 6-up RBDs, 12 scFvs, 6.11 Å. Two perpendicular views of Omicron S–FD01 are shown in surface representation, with 16L9 ScFv in lime and GW01 ScFv in cornflower blue.

- 3 Wang, Y. *et al.* Combating the SARS-CoV-2 Omicron (BA.1) and BA.2 with potent bispecific antibodies engineered from non-Omicron neutralizing antibodies. *Cell Discov* **8**, 104, doi:10.1038/s41421-022-00463-6 (2022).

10i. The modified manuscript claims that the mAb should work for all variants based on GSAID data and 95% conservation on a limited set of positions. Is that true across all of the PISA positions? Please show a table with the conservation at each position or a heat map on the RBD for the PISA positions. This will visually show this better and clarify the probability of

conservation across all contacts instead of a limited number.

I also appreciate the modified sentence. However, even though they are independent, it is probably more accurate to say “two complementary epitopes” rather than “two highly conserved epitopes”.

Thank you for your valuable input. We have incorporated the conservation data at each position on the RBD into the PISA position in Supplementary Table S6. Additionally, we have updated the conservation data in Fig. 5d and 5e, noting that the conservation data is based on the sequence data from GISAID during 2023. Regarding the description of the G7-Fc epitope, we have changed “two highly conserved epitopes” to “two complementary epitopes” in the manuscript accordingly.

Supplementary Table S6. Conservation of the hydrogen bonding sites between XBB S-RBD and G7-Fc.

G7-Fc interaction	XBB S-RBD	Conservation (%)
7F3	LEU 455	83.03
	PHE 456	72.13
	TYR 473	95.44
	LYS 478	81.1
	GLY 485	94.9
	ASN 487	95.68
	CYS 488	95.74
	TYR 489	95.66
	SER 490	68.84
	PRO 491	95.67
	GLN 493	95.61
	SER 494	95.03
	TYR 501	94.9
GW01	PHE 375	91.79
	PHE 377	92.59
	LYS 378	92.62
	ASN 405	93.29

14. Thank you for the response.

Models of synergy for antibodies include Zhou (Science. 2022. PMID:35324257) which proposed a 2-step models for antibody synergy where binding of the first antibody locks RBD into the up position and enhances the binding of the second antibody. For multispecifics, the Zhou model and crosslinking have been observed (Sci Transl Med. 2021. PMID:34519517 and bioRxiv. 2022. PMID:35982683)). This paper proposes synergy is due to increase surface area from GW01 binding. But given the crosslinking of spikes observed in Cryo-EM, this is likely a major mechanism and should be investigated. To determine the relative role of crosslinking to the increased potency observed vs the role of GW01 binding in the bispecific, binding and neutralization of a G7-Fc configured molecule where the GW01 component was replaced with a genetically comparable scFv that targets a different virus should be made and tested.

This would discriminate between the two possible hypotheses and should be performed. Once the mechanism is determined, it should be compared to that of published models for synergy for monoclonals and multispecifics.

We agree that G7-Fc also use the “2-step models for antibody synergy where binding of the first antibody locks RBD into the up position and enhances the binding of the second antibody”. 7F3 binding would induce RBD into the up position and enhances the binding of GW01. We claim the “larger interface area” helped the binding because mixing two antibodies would not help. Only the crosslinked GW01-7F3 can bind to S trimer. We added a sentence to describe this in Line 348-349: “7F3 binding might induced RBD to the up position which allows GW01 binding.”

As you suggested, we replaced GW01 scFv with 4L12 or REGN10989 and mixed these two bispecific antibodies with XBB S. Negative staining EM images showed that incubation of 4L12-7F3-Fc or REGN10989-7F3-Fc with S did not induce the formation of a trimer dimer consisting of two S trimers. This indicated that two scFvs of G7-Fc synergistically induced the crosslink of XBB S trimers. We added the negative staining EM images in Supplementary Fig. S4c and revised the statement as follows in line 337-341:

“In addition, the trimer dimer state of spike was not induced by incubating 4L12-7F3-Fc or REGN10989-7F3-Fc with XBB S, as seen in negative staining EM images (Supplementary Fig. S4c). These results demonstrate that 7F3 and GW01 work synergistically, inducing the crosslink of spike and improving binding and neutralizing activity with the RBD.”

Supplementary Fig. S4 (c) Negative staining EM images of SARS-CoV-2 XBB S alone, XBB S/G7-Fc complex, XBB S/G7-Fab, XBB S/7F3 IgG/GW01 IgG, XBB S/7F3 IgG, XBB S/GW01 IgG, XBB S/4L12-7F3-Fc and XBB S/REGN10989-7F3-Fc showing that G7-Fc binding induces the formation of trimer dimer.

16. Thank you for adding the method. It is unusual to use area under the curve to calculate percent competition. Most groups use maximum binding of the analyte relative to its expected binding. Please add the similar calculation method the antibody competition experiments.

Thank you for your diligent review and valuable feedback. We have now updated the method in the manuscript to calculate the percent competition using the maximum binding instead of the area under the curve.

Other comments section numbering

1. Thank you for modifying the text. MERS-CoV has not caused a pandemic.

Thank you for your comment. We have revised the sentence as follows: *"Moreover, considering that coronaviruses have incited two pandemics and one epidemic within the past two decades, involving SARS-CoV, SARS-CoV-2, and MERS-CoV, the demand for pan-sarbecovirus antibodies becomes paramount in anticipation of potential recurrences of coronavirus pandemics."*

We acknowledge and value your meticulous attention to the details, affirming our commitment to upholding the precision and dependability of our research outcomes.

REVIEWER COMMENTS

Reviewer #3 (Remarks to the Author):

Thank you for the responses. I still have the following concerns and comments:

10b. There are several issues with the response and figure. The response indicates that following GW01 Fab binding, there is not 7F3 binding. Similarly, following 7F3 binding there is no GW01 binding. However, in the new sup Fig 11 a and b, there is a binding signal shown for the later step in both panels (the boxed in sections). The right-hand curves of Sup 11b does not match the boxed area to it's left. Also, I am confused because GW01 is not supposed to bind and neutralize XBB, so why is there high binding of GW01 Fab? Furthermore, Sup Fig 11b left panel exactly matches Sup Fig 11a left when overlaid (except for the colors).

Is this supposed to be a schematic of what the data should look like? If so, please replace with the real data not the predicted. The response claims and figure data do not match and need to be rectified so that the data can be properly reviewed and interpreted.

In regard to the response relating to the formation of dimers with G7-Fab is used in EM does suggest the two arms of the IgG are required for crosslinking. However, the reviewer wanted (by EM or Octet) to answer if GW01 binding in the bispecific was due to the format or to the fact that 7F3 bound first and therefore allowed GW01 binding to XBB. The EM experiment proposed by the reviewer was to saturate binding of Fab 7F3 to XBB prior to adding Fab GW01 to see if GW01 would then be able to bind XBB.

14. (Line 376-384). Please include a statement about the 2-step synergy mechanism being similar to other proposed models and reference those publications.

Reviewer #3 (Remarks to the Author):

Thank you for the responses. I still have the following concerns and comments:

We are grateful for the reviewer's insightful comments and suggestions, which have enabled us to conduct a more comprehensive analysis of the data and significantly enhance the quality of our work.

10b. There are several issues with the response and figure. The response indicates that following GW01 Fab binding, there is not 7F3 binding. Similarly, following 7F3 binding there is no GW01 binding. However, in the new sup Fig 11 a and b, there is a binding signal shown for the later step in both panels (the boxed in sections). The right-hand curves of Sup 11b does not match the boxed area to it's left. Also, I am confused because GW01 is not supposed to bind and neutralize XBB, so why is there high binding of GW01 Fab? Furthermore, Sup Fig 11b left panel exactly matches Sup Fig 11a left when overlaid (except for the colors).

Is this supposed to be a schematic of what the data should look like? If so, please replace with the real data not the predicted. The response claims and figure data do not match and need to be rectified so that the data can be properly reviewed and interpreted.

In regard to the response relating to the formation of dimers with G7-Fab is used in EM does suggest the two arms of the IgG are required for crosslinking. However, the reviewer wanted (by EM or Octet) to answer if GW01 binding in the bispecific was due to the format or to the fact that 7F3 bound first and therefore allowed GW01 binding to XBB. The EM experiment proposed by the reviewer was to saturate binding of Fab 7F3 to XBB prior to adding Fab GW01 to see if GW01 would then be able to bind XBB.

We appreciate the reviewer's feedback and apologize for any confusion caused. In the previous version, Figure 11a depicted the experimental steps rather than presenting the actual binding curve results. We have conducted the experiment again and updated Supplementary Figure 11 with the real BLI data. The revised description in the manuscript now accurately reflects the updated results as follows in page 14:

“Besides, we examined the binding of the XBB trimer with both 7F3 Fab and GW01 Fab. The XBB S trimer immobilized onto the biosensor was first incubated with 7F3 Fab, followed by the addition of GW01 Fab (as illustrated by the red line in Supplementary Figure 11a). Minimal binding of 7F3 Fab was observed, and subsequent addition of GW01 Fab did not result in GW01 binding. Conversely, when GW01 Fab was first exposed to XBB trimer, no binding occurred, and subsequent addition of 7F3 Fab also failed to bind (depicted by the pink line in Supplementary Figure 11b).”

Additionally, XBB S trimers failed to form trimer dimers when incubated with G7-Fab (Supplementary Fig. S4b-c). These results indicate the critical role of the IgG format of the bispecific antibody in promoting spike crosslinking and enabling synergistic binding of 7F3 and GW01 to the XBB S trimers.”

Supplementary Fig. S11. Binding of XBB trimer with 7F3 Fab and GW01 Fab as measured by BLI. The NTA sensor tip, immobilized with XBB protein, was initially exposed to either the 1st Fab or the VRC01 isotype control for binding, followed by the binding of the 2nd Fab. **(a)** Binding of GW01 Fab to the XBB trimer subsequent to XBB binding with 7F3 Fab. The red line indicates the sensor loaded with XBB trimer binding to 7F3 Fab first and then to GW01 Fab. The green line signifies the sensor

loaded with XBB trimer binding to the isotype control in both steps. The zoomed-in view of XBB binding with the 1st Fab and 2nd Fab is depicted above. **(b)** Binding of 7F3 Fab to the XBB trimer subsequent to XBB binding with GW01 Fab. The pink line indicates the sensor loaded with XBB trimer binding to GW01 Fab first and then to 7F3 Fab.

14. (Line 376-384). Please include a statement about the 2-step synergy mechanism being similar to other proposed models and reference those publications.

Thanks for your suggestion. We added more details to discuss the 2-step and crosslinking mechanism and include the references in the discussion (page 14):

“Structural analysis of XBB S-G7-Fc complex showed that GW01 could only binds to up RBD and would clash with the adjacent down RBD (Supplementary Fig. S7a). Thus, 7F3 binding might induced RBD to the up position, thus facilitating GW01 binding. This resembles the 2-step models for antibody synergy^{25,30}, where binding of the first antibody induces RBD into the up position and enhances the binding of the second antibody. However, neither 7F3 nor GW01 alone, nor their mixture, could bind to XBB S, suggesting that the 2-step model alone cannot fully elucidate this. It is necessary to link 7F3 and GW01 to enable the interaction between XBB S and G7-Fc. The structural organization of the bispecific antibody, particularly the Fc region, is crucial in enabling G7-Fc to bind to the S trimer and initiate the formation of trimer dimers.”

25 Wang, Y. *et al.* Combating the SARS-CoV-2 Omicron (BA.1) and BA.2 with potent bispecific antibodies engineered from non-Omicron neutralizing antibodies. *Cell discovery* **8**, 104, doi:10.1038/s41421-022-00463-6 (2022).

30 Zhou, T. *et al.* Structural basis for potent antibody neutralization of SARS-CoV-2 variants including B.1.1.529. *Science* **376**, eabn8897, doi:10.1126/science.abn8897 (2022).

REVIEWER COMMENTS

Reviewer #3 (Remarks to the Author):

Comments on 10b

Thank you for providing the explanation and full curves for the experiments described on page 14 and in supplemental figure 11. The binding of the XBB trimer shows a shift of about 2 (assume nm because it is BLI, needs unit label). However, in Supplemental Fig 11a, the binding of the 7F3 Fab is so low it needs to be zoomed in and is less than 0.05. This is very low and suggests that the concentration of 7F3 Fab used is too low. Furthermore, in Supplemental Figure 11b, one would expect to see 7F3 bind in the 2nd Fab binding step similarly to the binding seen in 11a 1st binding step. This is not seen and the 7F3 binding is negative and lower than the binding of the negative control VRC01. This is again suggestive that the concentration of Fab being used is too low.

Since 7F3 neutralizes XBB, one would expect to see much higher binding signal. That is, I would expect to see similar binding to the association curves shown in Supplemental figures S2, where XBB binding was ~0.1 to ~1.8 (by eye estimation). Even when accounting for the Fab having 1/3 of the mass of a full IgG, one would expect higher binding in this assay if the concentration chosen was sufficient. Additionally in Fig S2, the signals at the highest concentrations of 7F3 are not saturated, suggesting that the binding signal could be even higher at saturation in Supp Figure 11a.

These issues are important for several reasons. First, because the binding of 7F3 Fab is so low, it is hard to determine if 300 seconds was sufficient to achieve saturated binding or if the curve would continue gradually going up. It appears to be going gradually up. Second, in this experiment, GW01 binding signal would be dependent on 7F3 binding opening up the RBD to allow GW01 to bind. Therefore, it is reasonable to expect that GW01 Fab binding to XBB might occur at a lower affinity and/or association rate and therefore have a lower signal than 7F3. Thus, the extremely low maximum binding of 7F3 Fab may significantly lower the sensitivity of seeing GW01 Fab binding in this assay.

Taken together the data in this experiment does not support the results and conclusions that are being made from it.

Finally, apologies if I missed it, but I do not see concentration or methods for this experiment.

To address my concerns, the authors would need to do one of the following things:

1) If they would like to make the claims on lines 338-348 and the associated portions of the discussion (385-387), they should repeat the experiment shown in Figure 11a and 11b, where the binding of 7F3 is both higher and saturated. Specifically, in panel a “1st Fab binding” the binding signal should be above 0.5 nm shift (ideally near 1 nm) and should be of sufficient time that saturation is observed as a plateau of the curve. This should be achievable because 7F3 binds to XBB trimer as an IgG in Figure S2 with a maximal shift of close to 5 nm. In addition, the same concentration of Fab GW01 and time should be used for the “2nd Fab binding” step in panel 11a.

Next, assuming Fab GW01 does not show binding in panel 11a and that the same concentrations of Fabs are used in panel 11b, panel 11b should show similar Fab 7F3 binding in the “2nd Fab binding” step as it has in panel 11a “1st Fab binding” step. This is because if Fab GW01 does not binding XBB trimer, there should not be any competition limiting the ability of Fab 7F3 to bind in panel 11b’s “2nd Fab binding” step.

Lastly, the authors will need to provide a method for the supplemental figure 11 in the methods section. This method should include the sensor type used, buffers used, concentrations of XBB trimer and the concentrations of the Fabs.

2) If the authors do not want to repeat the experiment, then they should remove the claim relating to the IgG format being critical to synergistic binding and remove Figure S11. However, I think it would be reasonable to propose this hypothesized mechanism as long as they also mention alternative hypotheses such as the hypothesis that the experiment in supplemental figure S11 was seeking to rule out.

Comments on 14. No further concerns on this.

Reviewer #3 (Remarks to the Author):

Comments on 10b

Thank you for providing the explanation and full curves for the experiments described on page 14 and in supplemental figure 11. The binding of the XBB trimer shows a shift of about 2 (assume nm because it is BLI, needs unit label). However, in Supplemental Fig 11a, the binding of the 7F3 Fab is so low it needs to be zoomed in and is less than 0.05. This is very low and suggests that the concentration of 7F3 Fab used is too low. Furthermore, in Supplemental Figure 11b, one would expect to see 7F3 bind in the 2nd Fab binding step similarly to the binding seen in 11a 1st binding step. This is not seen and the 7F3 binding is negative and lower than the binding of the negative control VRC01. This is again suggestive that the concentration of Fab being used is too low.

Since 7F3 neutralizes XBB, one would expect to see much higher binding signal. That is, I would expect to see similar binding to the association curves shown in Supplemental figures S2, where XBB binding was ~0.1 to ~1.8 (by eye estimation). Even when accounting for the Fab having 1/3 of the mass of a full IgG, one would expect higher binding in this assay if the concentration chosen was sufficient. Additionally in Fig S2, the signals at the highest concentrations of 7F3 are not saturated, suggesting that the binding signal could be even higher at saturation in Supp Figure 11a.

These issues are important for several reasons. First, because the binding of 7F3 Fab is so low, it is hard to determine if 300 seconds was sufficient to achieve saturated binding or if the curve would continue gradually going up. It appears to be going gradually up. Second, in this experiment, GW01 binding signal would be dependent on 7F3 binding opening up the RBD to allow GW01 to bind. Therefore, it is reasonable to expect that GW01 Fab binding to XBB might occur at a lower affinity and/or association rate and therefore have a lower signal than 7F3. Thus, the extremely low maximum binding of 7F3 Fab may significantly lower the sensitivity of seeing GW01 Fab binding in this assay.

Taken together the data in this experiment does not support the results and conclusions that are being made from it.

Finally, apologies if I missed it, but I do not see concentration or methods for this experiment.

To address my concerns, the authors would need to do one of the following things:

- 1) If they would like to make the claims on lines 338-348 and the associated portions of the discussion (385-387), they should repeat the experiment shown in Figure 11a and 11b, where the binding of 7F3 is both higher and saturated. Specifically, in panel a "1st Fab binding" the binding signal should be above 0.5 nm shift (ideally near 1 nm) and should be of sufficient time that saturation is observed as a plateau of the curve. This should be achievable because 7F3 binds to XBB trimer as an IgG in Figure S2 with a maximal shift of close to 5 nm. In addition, the same concentration of Fab GW01 and time should be used for the "2nd Fab binding" step in panel 11a.

Next, assuming Fab GW01 does not show binding in panel 11a and that the same concentrations of Fabs are used in panel 11b, panel 11b should show similar Fab 7F3 binding in the "2nd Fab binding" step as it has in panel 11a "1st Fab binding" step. This is because if Fab GW01 does not bind XBB trimer, there should not be any competition limiting the ability of Fab 7F3 to bind in panel 11b's "2nd Fab binding" step.

In response to the reviewer's suggestion, we adjusted the Fab concentration in the **Supplementary Figure 11**. The starting concentration of mAb in Supplementary Figure S2 was 300 nM. To maintain comparability between the experiments, the starting concentration of a Fab was 600 nM, considering that Fab is monovalent while IgG is bivalent with two Fabs. When the XBB S trimer was first incubated with 7F3 Fab followed by GW01 Fab (green line in **Figure 1**), low binding of 7F3 Fab (about 0.1) was observed, and subsequent addition of GW01 Fab showed minimal binding (less than 0.05). Conversely, when GW01 Fab was first exposed to XBB trimer, minimal binding of GW01 was observed (less than 0.05), and subsequent addition of 7F3 Fab also showed very low binding to the XBB trimer (about 0.1, orange line in **Figure 1**). The continued minimal binding of 7F3 Fab to the XBB trimer, even after increasing the Fab concentration, was surprising. We agree with the reviewer's assessment that 7F3 effectively neutralizes XBB, as demonstrated by the robust binding of 7F3 IgG to the RBD of XBB in **Supplementary Figure S2**. We hypothesize that the functionality of 7F3 Fab is compromised upon removal of the Fc region.

Figure 1 Binding of XBB trimer to 7F3 Fab and GW01 Fab as measured by BLI. The NTA sensor tip, immobilized with XBB protein, was initially exposed to either the 1st Fab for binding, followed by the binding of the 2nd Fab. The zoomed-in view of XBB binding with the 1st Fab and 2nd Fab is depicted above. Binding of GW01 Fab to the XBB trimer subsequent to XBB binding with 7F3 Fab (the green line). Binding of 7F3 Fab to the XBB trimer subsequent to XBB binding with GW01 Fab (the orange line). VRC01 Fab (one of HIV NAb) was used as isotype control. Binding data were collected using Octet BLI acquisition software 8.0 and analyzed with Octet® BLI analysis software 8.1, after subtracting the VRC01 Fab signal.

To further investigate the binding capability of 7F3 Fab, we first assessed its interaction with the WT RBD. As depicted in **Figure 2a**, both GW01 and its Fab form demonstrated strong binding to

the WT RBD, indicated by a binding signal shift of 3 to 4. In contrast, 7F3 IgG showed pronounced binding to the WT RBD, whereas the binding of 7F3 Fab to WT RBD was significantly reduced. 7F3 bound less to the XBB trimer (**Figure 2b**) than to the XBB RBD (**Supplementary Figure S2**), with values around 0.2. 7F3 Fab had even lower binding, at about 0.1 in **Figure 2b**. These results suggest that the binding activity of 7F3 Fab is notably impaired upon removal of the Fc region, emphasizing the critical role of the Fc region for 7F3's interaction with the WT RBD.

We then compared the binding of the XBB S trimer to G7-Fc and a combination of 7F3 Fab and GW01 Fab. To maintain comparability between the experiments, the starting concentration of a Fab was 600 nM. XBB trimer exhibited a high binding affinity to G7-Fc with a K_D value of 1.39 nM (**Figure 2c**). In contrast, XBB trimer exhibited no detectable binding affinity to the combination of 7F3 Fab and GW01 Fab (**Figure 2d**). This result further confirms that the binding activity of 7F3 Fab in G7-Fc is significantly impaired upon removal of the Fc region.

Based on the results of **Figure 2**, 7F3 stands out as a unique antibody; the removal of its Fc region notably diminishes its binding capacity to the XBB trimer. Thus, the 7F3 Fab data in Supplementary Figure 11 may not accurately depict 7F3's true functionality. It's conceivable that G7-Fc operates via a 2-step model for antibody synergy. To prevent potential misconceptions, we have opted to exclude Supplementary Figure 11, along with its related results and conclusions (338-348), as well as relevant sections of the discussion (385-387) from the manuscript.

Figure 2 Binding of WT RBD and XBB trimer to 7F3 Fab and GW01 Fab as measured by BLI. Binding of (a) WT RBD and (b) XBB trimer to 7F3 Fab and GW01 Fab. 7F3 and GW01 IgG were used as controls. Binding of XBB trimer to (c) G7-Fc and (d) the combination of 7F3 Fab and GW01 Fab.

Lastly, the authors will need to provide a method for the supplemental figure 11 in the methods section. This method should include the sensor type used, buffers used, concentrations of XBB trimer and the concentrations of the Fabs.

Below are the methods related to Figure 1 in this response and the supplemental figure 11:

Preparation of Fab for BLI

We generated Fab fragments by digesting IgG using immobilized papain (Thermo Scientific). Initially, 2 mg of IgG was dissolved in 0.5 mL of sample buffer (20 mM sodium phosphate, 10 mM EDTA, pH 7.0). Concurrently, 0.5 mL of immobilized papain was equilibrated in 4 mL of digestion buffer (20 mM Cysteine•HCl, 20 mM sodium phosphate, 10 mM EDTA, pH 7.0) twice. The IgG sample was then mixed with the immobilized papain and incubated at 37°C for 4 hours. Following papain digestion, 10 mM Tris•HCl was added to the digest. The immobilized papain was subsequently separated by centrifugation. Finally, Fab and Fc fragments were isolated from the supernatant using Protein A Resin (Gen Script).

Binding of XBB S trimer to 7F3 Fab and GW01 Fab by BLI

To assess the binding affinity of G7-Fc or the mixture of GW01 Fab and 7F3 Fab to XBB S trimer, we performed BLI experiments using the Octet RED96e System. Both XBB S and antibody samples were diluted in PBST buffer (0.05% Tween-20 in PBS) for this BLI experiment. For the sequential binding assay, the sensors loaded with XBB S trimer were immersed in the first Fab at a concentration of 600 nM for 300 seconds. Following a baseline stabilization in PBST for 120 seconds, the sensors were exposed to the second Fab at 600 nM to assess its binding. VRC01 Fab (one of HIV NAb) was used as isotype control. Binding data were collected using Octet BLI acquisition software 8.0 and analyzed with Octet® BLI analysis software 8.1, after subtracting the VRC01 Fab signal.

We did not include these methods in the revise manuscript because we removed Figure S11.

2) If the authors do not want to repeat the experiment, then they should remove the claim relating to the IgG format being critical to synergistic binding and remove Figure S11. However, I think it would be reasonable to propose this hypothesized mechanism as long as they also mention alternative hypotheses such as the hypothesis that the experiment in supplemental figure S11 was seeking to rule out.

We repeated the experiments from **Supplementary Figure 11** and conducted additional tests to the best of our ability. Our findings indicate that **Supplementary Figure 11** does not accurately depict the true functionality of 7F3 when integrated into the G7-Fc antibody. Consequently, we have removed Figure S11, along with its related conclusions and discussions.

We appreciate the reviewer's invaluable suggestions for enhancing the manuscript. We trust that this revised version addresses the concerns raised and improves the overall quality of the paper.